# INFORMATION-THEORETIC UNDERPINNINGS OF GENERALIZATION AND TRANSLATABILITY IN EMERGENT COMMUNICATION

## ABSTRACT

Traditional emergent communication (EC) methods often fail to generalize to novel settings or align with representations of natural language. Here, we show how controlling the Information Bottleneck (IB) tradeoff between complexity and informativeness (a principle thought to guide human languages) helps address both of these problems in EC. Using VQ-VIB, a recent method for training agents while controlling the IB tradeoff, we find that: (1) increasing pressure for informativeness, which encourages agents to develop a shared understanding beyond task-specific needs, leads to better generalization to more challenging tasks and novel inputs; (2) VQ-VIB agents develop an EC space that encodes some semantic similarities and facilitates open-domain communication, similar to word embeddings in natural language; and (3) when translating between English and EC, greater complexity leads to improved performance of teams of simulated English speakers and trained VQ-VIB listeners, but only up to a threshold corresponding to the English complexity. These results indicate the importance of informational constraints for improving self-play performance and human-agent interaction.

## 1 INTRODUCTION

We wish to develop artificial agents that communicate in grounded settings, via communication that enables high task utility, generalizability to novel settings, and good human-agent cooperation. Emergent communication (EC) methods, wherein agents learn to communicate with each other in an unsupervised manner by maximizing a reward function, take a step towards this vision by producing agents that use grounded communication (Lowe et al., 2017; 2020; Lazaridou & Baroni, 2020). While numerous EC methods have succeeded in training agents to communicate with each other to solve a particular task, they still fall short of the vision of generalizable and human-interpretable communication. For example, agents trained to discriminate between two types of images will fail to discriminate between sixteen images (Chaabouni et al., 2021b), and messages often violate human expectations for meanings (Kottur et al., 2017).

In this work, we take steps towards addressing these limitations by building on the information-theoretic EC approach of Tucker et al. (2022). This approach connects EC with the Information-Bottleneck (IB, Tishby et al., 1999) framework for semantic systems (Zaslavsky et al., 2018; Zaslavsky, 2020), via the vector-quantized variational Information Bottleneck (VQ-VIB) neural architecture (Tucker et al., 2022). Specifically, VQ-VIB agents are trained to optimize a tradeoff between maximizing utility (how well they perform a task), maximizing informativeness (how well a listener can infer a speaker's meaning, independently of any downstream task), and minimizing communicative complexity (roughly the number of bits allocated for communication). While previous EC methods typically focus on task-specific utility maximization (Lowe et al., 2017), there is broad empirical evidence suggesting that human languages are guided by the IB informativeness-complexity tradeoff (Zaslavsky et al., 2018; 2019; 2021; 2022; Mollica et al., 2021). Therefore, we hypothesize that taking into account informativeness could improve EC generalizability to novel settings while adjusting complexity could improve the translatability between EC and human languages.

Results from our experiments support this hypothesis. First, we show that encouraging informativeness allows EC agents to generalize beyond their training distribution to handle more challenging

tasks and out-of-distribution objects, with VQ-VIB achieving the best performance compared to alternative EC methods. Second, we propose a simple method for translating natural language word embeddings (e.g., GloVe, Pennington et al., 2014) into EC signals and use that to simulate human-agent communication in a cooperative object-discrimination task. We find that team performance for English speakers and trained EC agents improves with the communicative complexity of the EC system, but only up to a certain threshold, which corresponds to the complexity of the English object naming system. Together, our findings suggest that training EC agents while controlling the informativeness-complexity tradeoff, in addition to maximizing utility, may simultaneously support improved self-play performance as well as human-agent interaction.

## 2  RELATED WORK

Our work builds upon prior research in emergent communication (EC), wherein cooperative agents are trained to maximize a reward function. For example, a speaker may observe a "target" image that a listener must identify from a pool of candidate images, based only on the speaker's emitted communication. Researchers in EC often consider the effects of different neural network architectures, training losses, or environmental factors on learned communication (Kottur et al., 2017; Mu & Goodman, 2021; Kuciński et al., 2021; Tucker et al., 2021). In this work, we are primarily concerned with how information-theoretic properties of agents' communication allow them to generalize or align with human languages, although we also improve upon an existing neural architecture.

### 2.1  GENERALIZATION OF EMERGENT COMMUNICATION

Several works consider the ability of EC agents to generalize to different settings than those in which they were trained. For example, Lazaridou et al. (2018) train agents with symbolic inputs and use novel combinations of such inputs to test generalization. Other researchers use similar train-test gaps to measure the compositionality and generalizability of communication, but such experiments are necessarily restricted to domains with symbolic structure (as those allow recombination of features) (Andreas, 2019; Kuciński et al., 2021; Spilsbury & Ilin, 2022). Conversely, Chaabouni et al. (2021b) tests agents' abilities to generalize in image-based reference games to 1) more distractor images or 2) inputs from a distinct dataset of images. They find that using harder tasks at training time is important for improving test-time performance and that using population-based voting improves cross-domain transfer. We are similarly interested in the ability of agents to generalize to harder tasks and novel inputs, but we focus on training only a single team of agents; research in learning dynamics due to population effects or changes in environment is complementary to our work.

### 2.2  LINKS TO NATURAL LANGUAGE

Complementing research of EC in self-play (evaluating agents that were trained together), many researchers explore connections between EC and natural language. Some top-down methods combine pretrained language models with finetuning in grounded environments, but such methods can suffer from "drift" wherein agents learn to ascribe new meanings to words (Lewis et al., 2017; Lazaridou et al., 2020; Jacob et al., 2021). Other researchers train agents in self-play and then seek to connect learned EC to human-interpretable concepts or natural language (Andreas et al., 2017; Kottur et al., 2017). Tucker et al. (2021) take a middle ground by training agents to communicate via discrete representations in a continuous space, which can be linked to natural word embeddings via supervision during training. We seek to uncover methods for best connecting EC and human communication via translation, with no supervision data when training the agents. Based on information-theoretic analysis of human naming systems, we investigate whether producing EC that matches humans' complexity and informativeness enables better translation.

## 3  BACKGROUND: INFORMATION-THEORETIC EMERGENT COMMUNICATION

Our work builds on the information-theoretic framework of Zaslavsky et al. (2018) for semantic systems, and especially on its extension to scalable emergent communication in artificial neural agents, proposed by Tucker et al. (2022). Below we review the theoretical and empirical foundations of this line of work.

### 3.1 EFFICIENT COMPRESSION AND SEMANTIC SYSTEMS

Zaslavsky et al. (2018) argued that languages are shaped by the need of speakers and listeners to efficiently compress meanings into words. They formulated this objective using the Information Bottleneck (IB, Tishby et al., 1999) principle, which can be interpreted as a tradeoff between the informativeness and complexity of the lexicon, and can be derived from rate-distortion theory (Shannon, 1959; Harremoës & Tishby, 2007). In this framework, a speaker is characterized as a probabilistic encoder $S(c|m)$ that, given an input meaning $m \sim p(m)$, generates a communication signal, $c$. A listener is characterized as a probabilistic decoder $D(\hat{m}|c)$, that reconstructs the speaker's meaning from $c$. Given such agents, complexity is computed as the mutual information between the speaker's meanings and signals, $I(m; c)$, which is roughly the number of bits that are need to encode $m$ using $c$. Informativeness measures how well the listener's interpretation, $\hat{m}$, matches the speaker's intended meaning; intuitively, this corresponds to how well a listener is able to understand a speaker. The agents' meanings, $m$ and $\hat{m}$, are defined as belief states, i.e., probability distributions over a feature vector representing the agents' environment. Therefore, maximizing informativeness is equivalent to minimizing the expected Kullback-Leibler (KL) divergence between the agents' belief states, $\mathbb{E}[D_{KL}[m\|\hat{m}]]$. Minimizing complexity yields very simple but uninformative communication systems, while maximizing informativeness may yield very complex systems. Optimal speakers and listeners therefore jointly balance the tradeoff between these two objectives by minimizing $I(m; c) - \beta\mathbb{E}[D_{KL}[m\|\hat{m}]]$, where $\beta \geq 0$ controls the tradeoff. This optimization problem is equivalent to the IB principle.

This theoretical framework has gained broad empirical support across human languages and multiple semantic domains, including color (Zaslavsky et al., 2018; 2022), artifacts (Zaslavsky et al., 2019), and personal pronouns (Zaslavsky et al., 2021). Specifically, it was found that the ways languages map meanings into words achieve near-optimal compression in the IB sense, suggesting that the same principle may also apply to communication among artificial agents (Zaslavsky et al., 2017; Eisape et al., 2020; Chaabouni et al., 2021a).

### 3.2 VECTOR-QUANTIZED VARIATIONAL INFORMATION BOTTLENECK (VQ-VIB)

Directly solving the IB optimization problem for high-dimensional inputs and neural agents is challenging, and in many emergent communication settings intractable. Tucker et al. (2022) proposed vector-quantized variational Information Bottleneck (VQ-VIB), a deep learning method that combines notions from vector quantization variational autoencoders (VQ-VAE, Van Den Oord et al., 2017) and variational Information Bottleneck (VIB). In addition, Tucker et al. (2022) integrated the standard utility-maximization approach with IB and showed how to train EC agents by trading off utility, informativeness, and complexity.

To apply IB to EC, note that the communication setup from §3.1 can be applied to Lewis reference games (Lewis, 2008), as depicted in Figure 1. Here, $m$ (the speaker's internal representation) is produced by passing a target input, $x_t$, through a pretrained VAE to model perceptual noise. Thus, in our case $m \in \mathbb{R}^d$. A decoder, $D$, seeks to reconstruct $m$ given the speaker's communication, $c$. Using this reconstruction, a listener agent, $L$, attempts to identify $x_t$ from a set $\mathcal{C} = \{x_1, ..., x_C\}$ of (noisily observed) candidate inputs by selecting $y \in \mathcal{C}$. VQ-VIB agents fit within this framework:

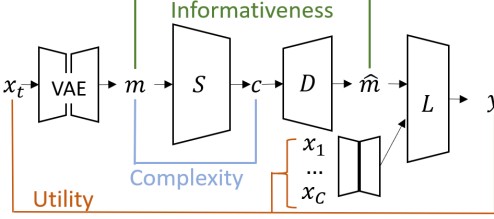

Figure 1: Communication setup in which agents coordinate to identify a target input (e.g., an image), $x_t$, among a set of candidates $\mathcal{C} = \{x_1, \ldots, x_C\}$ (see main text for details).

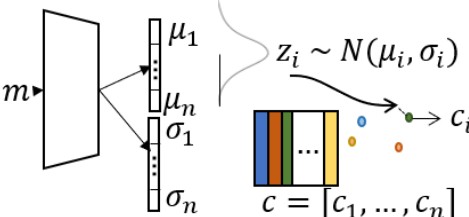

Figure 2: Our modified VQ-VIB architecture combines $n$ quantized vectors into a single message, enabling $k^n$ unique messages.

a probabilistic speaker, $S$, maps from $m$ to parameters of a Gaussian distribution, $\mu$ and $\sigma$, in a continuous latent space, $\mathbb{R}^Z$. A latent vector, $z$, is sampled from $\mathcal{N}(\mu(m), \sigma(m))$ and discretized by looking up the nearest element of a (trainable) codebook of $k$ quantization vectors $\zeta \in \mathbb{R}^Z$. The final communication vector output by the speaker, $c$, is this nearest quantization vector. That is, the VQ-VIB encoder is defined by $S(c|m) = \mathbb{P}(c = \text{argmin}_\zeta[\|z - \zeta\|^2]|m)$.

In this information-theoretic emergent communication (ITEC) framework, agents maximizes a combination of utility and informativeness while minimizing complexity, with $\lambda_U$, $\lambda_I$, and $\lambda_C$ controlling the relative weight of each term. In practice, agents are often trained with variational bounds for these terms, as shown in Equation 1 for VQ-VIB agents.

$$\text{maximize} \quad \lambda_U \mathbb{E}[U(x_t, y)] - \lambda_I \mathbb{E}[\|m - \hat{m}\|^2] - \lambda_C \mathbb{E}\left[D_{\text{KL}}[\mathcal{N}(\mu(m), \sigma(m))\|\mathcal{N}(0, 1)]\right] \quad (1)$$

Utility, $U(x_t, y)$, is task-specific, and in our settings it measures the listener's accuracy in identifying the target input. Informativeness is task-agnostic, measuring the listener's ability to reconstruct the speaker's belief state ($m$) given a communication vector $c$, regardless of any downstream task. This term may be approximated by $-\|m - \hat{m}\|^2$. Lastly, for VQ-VIB agents in particular, the Gaussian sampling in the latent space permits the upper bound of complexity as the KL divergence between $\mathcal{N}(\mu(m), \sigma(m))$ and a unit Gaussian (Alemi et al., 2017). Backpropagation through sampling is accomplished via the reparametrization trick (Kingma & Welling, 2013), and a straight-through estimator for the discretization process is used, as in VQ-VAE (Van Den Oord et al., 2017).

## 4 TECHNICAL APPROACH

We extend beyond prior work in two ways. First, we modify the VQ-VIB architecture to increase the effective vocabulary size of agents. As we later validated in experiments, this architectural change allows agents to learn much richer communication protocols. Second, we introduce a simple translation framework for analyzing similarities between aspects of natural language and emergent communication.

### 4.1 COMBINATORIAL CODEBOOK FOR LARGER VOCABULARY

While VQ-VIB agents represent an important step towards information-bounded emergent communication, the specific architecture, as proposed by Tucker et al. (2022) is limited in an important way: agents may only communicate via one of the quantized vectors in their codebook. In contrast, we propose a simple modification to the VQ-VIB architecture to allow agents to select $n \geq 1$ vectors from the codebook before concatenating them into a single message, as depicted in Figure 2. Given a meaning, $m$, a speaker computes $n$ means and variances $(\mu_i, \sigma_i)$ from which $n$ latent, continuous vectors $z_i$ are sampled, each in $\mathbb{R}^{Z/n}$. These continuous vectors are then discretized via the standard vector-quantization procedure of returning the nearest entry, using a codebook of size $k$. Lastly, the $n$ discrete vectors ($[c_1, ..., c_n]$) are concatenated into a single message, $c \in \mathbb{R}^Z$. (Concatenating multiple quantized vectors is used in Product Quantization literature (Jégou et al., 2011; Baevski et al., 2020), but without the Gaussian-based sampling that forms the variational backbone of VQ-VIB.)

While architecturally simple, this change over Tucker et al. (2022) increases the number of unique messages the speaker may emit from $k$ to $k^n$, without increasing the number of network parameters. Indeed, for $k = 1024; n = 4$, as used in some of our experiments, one would need a codebook with more than $10^{12}$ entries to achieve similar vocabulary size with $n = 1$. In experiments, we found that this change dramatically increased the informativeness of learned communication. Lastly, we note that even for $n > 1$, we may use the same variational upper bound on complexity by penalizing divergence from a Gaussian distribution.

### 4.2 HUMAN-AGENT TRANSLATION

One of our goals in this work is to quantitatively analyze similarities between EC signals and natural language as a function of the complexity of the EC system. In particular, we leverage the fact that VQ-VIB agents communicate via discrete signals that are embedded in a continuous space (see also Tucker et al., 2021), much like grounded word embeddings. We therefore study the similarity between the VQ-VIB EC embedding space and word embeddings from natural language.

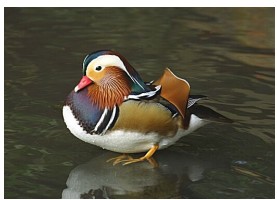 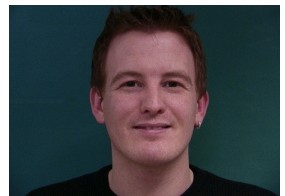 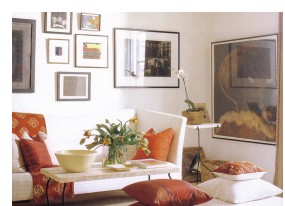

(a) `VG`: animals_plants;
Labels: **bird** (20), duck (14)

(b) `VG`: people;
Labels: **man** (24), person (2), human (1), male (1)

(c) `VG`: home;
Labels: **couch** (21), sofa (9)

Figure 3: Examples from the ManyNames dataset, with Visual Genome (`VG`) labels and English naming response (and counts). The `topname` of each image is shown in bold.

To this end, we propose a simple translation mechanism from natural language to EC. We assume access to a pre-trained neural speaker that outputs communication vectors, and to a dataset of images with natural language labels or descriptions. Natural language descriptions can be represented as real vectors using word-embedding methods, e.g., word2vec or GloVe (Mikolov et al., 2013; Pennington et al., 2014). One can construct a *translation dataset*, associating EC vectors and natural language embeddings, by (i) mapping each image to its label's word embedding, and (ii) passing the input image through an EC speaker to map it to an EC vector. Using this dataset, we train a least-squares linear model to map from natural language to EC. In general, non-linear translation models may be used as well; we leave exploration of such translators to future work.

In evaluation, we test the translatability of an EC system by combining a human speaker with a pre-trained EC listener. That is, given a target image, the human speaker generates a word which is represented by its embedding; that embedding is translated to an EC vector and passed to the EC listener, which then tries to pick the correct target image among all candidates. By evaluating the utility that this hybrid team achieves in this task we could assess the degree to which our EC agents are suited for communication and cooperation with humans.

## 5 EXPERIMENT DESIGN

We sought to measure the influence of complexity and informativeness on generalizability and translatability in EC. In experiments, we trained agents in self-play before evaluating them in novel settings or with simulated human partners.[1]

**Dataset**  We used the ManyNames dataset (Silberer et al., 2020), consisting of 25,000 images of natural objects with human-generated labels. This dataset is particularly useful in our context because it is composed of two parts (see Figure 3 for examples): (i) Visual Genome (`VG`) data, labeling each image with with one of seven high-level `VG` categories ['animals_plants', 'buildings', 'clothing', 'food', 'home', 'people', 'vehicles']; and (ii) free-naming data produced by approximately 36 native English speakers (the exact number varied due to filtering, see Silberer et al. (2020) for details), yielding a rich naming distribution of 1763 human-generated labels. These data naturally reflect the naming patterns and varying abstraction-levels that appear in English, as well as in other languages (e.g., Malt et al., 1999). Additionally, these phenomena are consistent with the Information Bottleneck view of object naming systems (Zaslavsky et al., 2019), that could settle on low-informativeness and low complexity (e.g., Figure 4a, `VG`) or higher informativeness and complexity (e.g., Figure 4a, `topname`). Given that these images elicited different levels of communicative complexity, we hoped that the dataset would similarly support a variety of EC patterns.

**Training setup**  We trained agents in the Lewis signalling game depicted in Figure 1: a speaker observed a "target" image and produced communication while a listener observed $C$ candidate images, including the target image, and the speaker's communication and had to select the target image from the candidates. We used a pretrained ResNet18 to extract 512 dimensional features from all images before passing them to the agents (He et al., 2016). In all experiments, we trained agents in self-play with no human supervision data. Agents were trained with a fixed number of candi-

---

[1]Code for experiments available at https://anonymous.4open.science/r/iclr_vqvib-B185

date images (typically, 2, unless using a baseline inspired by Chaabouni et al. (2021b)) drawn from distinct categories. In some experiments, we trained agents with images from separate `VG` classes; at other times we used distinct `topname` labels. By training agents with candidates from distinct sets, we intentionally separated the utility and informativeness terms. Note that this differs from experiments by Tucker et al. (2022), who did not use distinct candidates, so their informativeness and utility terms were maximized by the same policy.

**Evaluation in novel settings and domains**   At test time, we evaluated team utility (accuracy in identifying the target image) for various $C$, without enforcing that candidate images were drawn from distinct classes. Both changes made the task more challenging: increasing $C$ forced listeners to identify the target among more distractors, while allowing candidates from the same class allowed distractors to be more similar to the target. In some experiments, we also tested agents' ability to generalize to new image types. In such cases, we split the ManyNames dataset into distinct training and testing sets, without overlapping image domains. To this end, for each image, we recorded the most common label (of the 36 responses), referred to as `topname` (see Figure 3 for examples). We then randomly selected 20% of the 442 `topnames` in the entire ManyNames dataset and used all images with these `topnames` in the training set. The test set was generated by iterating over the remaining images in the dataset and retaining those for which no response (not merely that image's `topname`) matched a training set `topname`. For example, if the image of the duck in Figure 3 a were in the training dataset, no images with any responses for "bird" were included in the test set. We found that this train-test split procedure created sets of similar sizes that were substantially semantically different.

**Human-agent cooperation**   In our human-agent experiments, we tested the listener agent's ability to partner with human speakers. To this end, we used the free-naming English data to simulate a human speaker with the agents from the EC teams trained in earlier self-play experiments with distinct `topname` categories. We generated a "translation dataset" by randomly sampling 100 images; the `topnames` of the images were converted to GloVe embeddings (Pennington et al., 2014) and the speaker agent produced corresponding EC vectors. We then fit a linear translation model to map from GloVe embeddings to EC vectors; at test time, we measured the team performance when using the simulated English speaker, the translator, and the EC listener. To overcome random effects due to sampling so few images, during evaluation we performed 10 random trials for differently-trained translators and reported the mean effect.

**Agent variations**   To evaluate the influence of the informativeness–complexity tradeoff on generalization and translatability in EC, we varied $\lambda_I$, the weight for informativeness in Equation (1), while fixing $\lambda_U = 1$ and $\lambda_C = 0.01$. Increasing $\lambda_I$ increases the pressure for informative communication beyond any specific tasks, which in turn may increase the complexity of the communication system. In addition, we varied $n$, the number of quantized vectors per message, because (as demonstrated in experiments) we found that such architectural changes had large effects on the informativeness of learned communication. As baselines for evaluating our adjusted VQ-VIB agent architecture, we considered both onehot- and prototype-based architectures (Tucker et al., 2021; Chaabouni et al., 2021b). Details of agent architectures and hyperparameters are provided in Appendix A. In all experiments, we trained 5 agents from scratch for random seeds (0 to 4).

## 6   RESULTS

Here, we focus on the results for the VQ-VIB agents. Comparisons to other architectures (reported in Appendix C) corroborate Tucker et al. (2022)'s findings that VQ-VIB outperforms alternative EC methods (see Appendix C, Table 2 and Figure 10).

### 6.1   BASIC REFERENCE GAME

First, we trained agents in a simple reference game, with only two candidate images (the target and a distractor; $C = 2$) that were always drawn from distinct `VG` categories. This intentionally simple setting could be solved by learning a 7-element vocabulary (one for each `VG` domain) and correctly classifying each image. At test time, however, we evaluated agents in more challenging settings by allowing candidate images to belong to the same category (e.g., two vehicles) and by varying $C$ at test time (e.g., using 32 candidate images instead of 2). Results are shown in Figure 4. First,

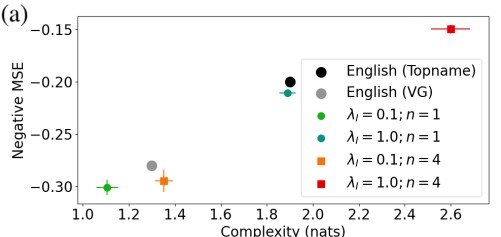
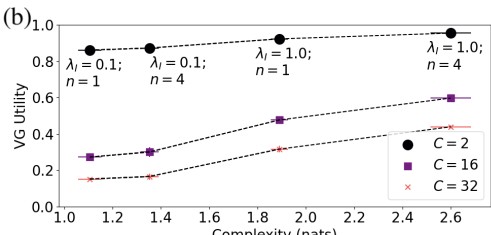

Figure 4: **Basic reference game.** (a) Measuring informativeness (as negative reconstruction MSE) vs. complexity shows the informativeness-complexity tradeoff characteristic of IB systems. (b) Team utility in the VG reference game, as a function of communication complexity. Using greater $\lambda_I$ and greater $n$ induced more complex communication, which improved utility, especially when evaluated with more candidates (different curves).

increasing $\lambda_I$ to encourage informativeness produces not only more complex and informative communication, as expected, but also greater utility. Second, increasing the combinatoriality parameter $n$ has a similar effect of increased complexity, presumably by allowing agents to learn to combine quantized vectors into more meaningful representations. Lastly, in this setting, greater complexity, whether achieved by varying $n$ or $\lambda_I$, supports greater team utility. Results using other speaker architectures and a greater range of $\lambda_I$ corroborate this trend, although VQ-VIB reached greater complexity and utility than other methods (Appendix C, Table 2).

## 6.2 GENERALIZING TO OUT-OF-DISTRIBUTION INPUTS

In our second suite of self-play experiments, we evaluated VQ-VIB on a more challenging generalization task: testing agents on out-of-distribution (OOD) image classes. We constructed train-test splits as explained in §5; agents were trained on 20% of the data and tested on images that did not share labels with the training set. As before, we trained agents with $C = 2$ and with images drawn from distinct categories (this time, distinctions by `topname` labels), but tested with varying numbers of candidates, without requiring candidates to be drawn from distinct categories. Thus, in our hardest evaluation settings, agents needed to simultaneously generalize to a larger number of candidates, drawn from non-distinct categories, which were not seen during training. Here, we present results for $n = 4$; see Figure 11 in Appendix D for other values of $n$.

Because informativeness, in contrast to utility, encourages agents to develop a shared understanding beyond specific task requirements, we expected that increasing $\lambda_I$ in training would improve the agents' ability to generalize to OOD inputs. Figure 5a confirms this intuition, showing that utility, as measured by the listener's accuracy in identifying the target image on OOD classes, increases with complexity as we vary $\lambda_I \in \{0, 0.1, 0.5, 1, 1.5, 2, 3, 10\}$ (as explained in §3, greater $\lambda_I$ yields greater complexity). ANOVA analysis finds that correlations were significant ($p < 0.05; t = 4.6, 8.7, 9.6$) for $C = 2, 16$, and 32, respectively. Note that each random trial used different agent initializations as well as different train-test splits. Interestingly, even as performance on OOD inputs increased with complexity, it increased slower than utility on in-distribution images, as seen in Figure 5b, which shows the mean difference between utility on the training and testing splits of the dataset. The growing generalization gap suggests a limit to OOD generalization, warranting further exploration in future research. However, even with this gap, VQ-VIB agents achieve substantially better OOD performance compared to onehot- and prototype-based agents (Figure 5 c for highlighted results and Appendix C for complete comparisons.)

Next, in order to gain more insight into the emergent semantic structure in VQ-VIB and how it may facilitate OOD generalization, we visualized the EC vectors from training and OOD images (Figure 6). To get a sense of the images associated with each EC vector, we iterated over the first 100 examples from an agent's training and testing set. Within each set, we recorded the speaker's output for each image and averaged the EC vectors across images with the same `topname`. We thus obtained the average EC vector for images of, e.g., "skater" or "steak." In general, we found that vectors that are close to each other often encode the same `topname`, i.e., they can be treated as synonyms, and therefore averaging simply reduced noise for visualization. Using 2D PCA (Jolliffe

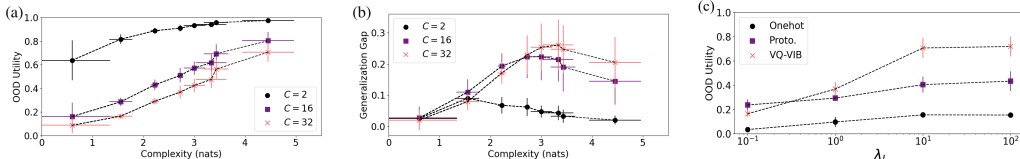

Figure 5: **Generalizing to out-of-domain inputs.** Listener accuracy as a function of complexity for OOD images improved with complexity (a) but the gap in performance between in- and out-of-distribution images grew (b). Compared to other architectures (for $C = 32$), VQ-VIB achieved greater OOD utility. Results generated for $n = 4$ for VQ-VIB while varying $\lambda_I$.

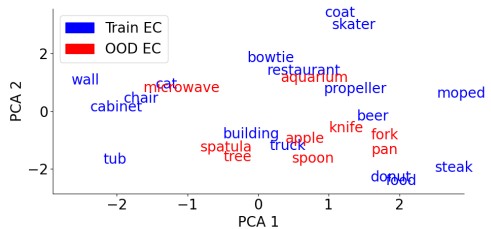

Figure 6: 2D PCA of EC vectors for training (blue) and OOD (red) images at complexity that roughly matches English (2.1 nats, $\lambda_I = 0.5$).

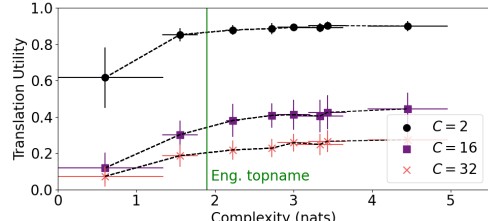

Figure 7: Performance of human-agent teams using GloVe to EC translation. Performance flattened after passing English complexity.

& Cadima, 2016), Figure 6 shows these vectors with their associated `topname` for images in the training set (blue) and the OOD test set (red). The agents learned a meaningfully structured semantic space, much like word embeddings in natural language. For example, in the training set, communication about food falls in one cluster, while furniture is in another. Remarkably, this semantic space also enables high performance on the OOD images because communication for held-out classes like "fork" or "microwave" can be interpreted based on its location in the pre-trained space. Visualization of the EC vectors at different complexities (Appendix B) indicates that VQ-VIB consistently forms structured semantic spaces, although the amount of information conveyed in that space varies.

Lastly, we tested whether using more candidates at training time enabled better OOD generalization. Chaabouni et al. (2021b) found that training with more candidates improved test-time performance, but they did not investigate the effect of varying $C$ on OOD inputs. Table 1 shows that increasing $\lambda_I$ led to greater improvements in OOD generalization compared to increasing the number of candidates $C$ at training time (although varying $C$ is more general than varying $\lambda_I$ in that it does not assume our information-theoretic framework). We believe this occurs because increasing $C$ at training time does not encourage agents to construct shared representations that are not task-specific, in contrast to increasing $\lambda_I$. Furthermore, increasing $C$ during training (for the same number of epochs) has a substantial overhead in runtime, while similar or even better performance can be achieved much faster by increasing $\lambda_I$ instead. For example, training with $C = 16$ and $\lambda_I = 1$ is about twice as fast as training with $C = 32$, while also achieving the best OOD performance.

### 6.3 ENGLISH-EC TRANSLATION

So far, we found that encouraging informative communication enables teams of VQ-VIB agents, operating in self-play, to better generalize to harder tasks and novel inputs. In this section, we studied the extent to which these agents can also usefully cooperate with humans by evaluating the translatability between English and EC. To this end, we measured team performance for a simulated English speaker and an EC listener by selecting random images from the ManyNames dataset, looking up the GloVe embedding for each image's `topname` (the most common name provided by the participants in the ManyNames English naming task), translating the embedding to EC via the linear translator (see §5), and then passing on the translated communication to the trained EC listener (as well as candidate images). Team utility, for varying agents and $C$, is plotted in Figure 7. At low complexity values, team performance exhibited similar trends to prior results: increasing complex-

Table 1: Mean (std. err.) utility evaluated on OOD images, for $n = 4$. Increasing $\lambda_I$ allowed better generalization to novel inputs, while increasing $C$ during training was less effective and slower.

| | | **Test C (out-of-distribution)** | | | | |
|---|---|---|---|---|---|---|
| **Train C** | $\lambda_I$ | 2 | 16 | 32 | Comp. (nats) | Time (min) |
| 2 | 0 | 0.64 (0.08) | 0.16 (0.58) | 0.09 (0.03) | 0.60 (0.37) | 17 (0.4) |
| | 1 | 0.91 (0.01) | 0.51 (0.04) | 0.37 (0.03) | 2.73 (0.03) | 16 (0.4) |
| 16 | 0 | 0.89 (0.02) | 0.38 (0.03 ) | 0.28 (0.02) | 2.04 (0.06) | 49 (2.1) |
| | 1 | **0.93 (0.01)** | **0.53 (0.03)** | **0.41 (0.02)** | 2.93 (0.08) | 50 (1.8) |
| 32 | 0 | 0.87 (0.02) | 0.37 (0.04) | 0.25 (0.02) | 1.98 (0.07) | 101 (8.1) |
| | 1 | 0.92 (0.02) | 0.48 (0.03) | 0.35 (0.02) | 2.88 (0.08) | 109 (2.7) |

ity and informativeness supported greater utility. However, at roughly 2 nats, team performance stopped improving even as EC complexity increased. This cutoff is particularly noteworthy because it nearly equals the complexity of the English naming system associated with this visual objects in the ManyNames dataset, calculated to be 1.9 nats using the MINE estimator (Belghazi et al., 2018) for the mutual information between English `topname` labels and images. A statistical likelihood ratio test of two nested linear models (one for utility vs. complexity, and another for a pair of linear models for utility vs. complexity, fitted separately for complexity greater or less than 2 nats) was significant ($p < 0.005$), confirming a plateauing behavior. Thus, we found that team utility is bottle-necked by the complexity of the English speaker, and that further increasing complexity in self-play affords no benefits. Appendix E includes similar results for translators fitted with more or less data; growing the dataset helped to some extent, but performance generally plateaued at 2 nats.

## 7  CONTRIBUTIONS

In this work, we explored the influence of informational constraints on two important properties of emergent communication (EC): generalization to harder tasks, including novel inputs, and transla-tion between EC and natural language. We suspected that most EC methods fall short of achieving these desired properties because they focus only on utility-maximization, without considering infor-mational constraints that are believed to shape human languages. We focused here on grounded meaning representations, and hypothesized that trading off informativeness and communication complexity, as suggested by the Information Bottleneck framework for semantic systems (Zaslavsky et al., 2018; Zaslavsky, 2020), would lead to EC systems that are better suited for human-like generalization and human-agent cooperation. To test this, we extended the VQ-VIB method for information-bounded EC agents (Tucker et al., 2022) to enable a richer and more scalable vocab-ulary structure. In addition, we proposed a simple translation mechanism between EC vectors and natural word embeddings, and used that to simulate human-agent communication in a cooperative image discrimination game.

Our results generally support our hypothesis and further show that VQ-VIB outperforms onehot- and prototype-based EC methods. In particular, we showed that: (1) encouraging informativeness allows agents to better generalize to more challenging tasks and out-of-distribution inputs; (2) the structure of the emergent VQ-VIB communication vectors encodes some semantic similarities and facilitates open-domain communication, similar to word embeddings in natural language; and (3) performance for teams of simulated English speakers and trained EC listeners improves with the complexity of the EC system, but only up to the complexity level of the English naming system. These results suggest that taking into account the IB informativeness-complexity tradeoff in EC, in addition to maximizing utility, may support both self-play performance and human-agent interaction.

Future work could build upon this framework in many directions. While we have shown the impor-tance of controlling informativeness and complexity in EC, further exploration of these constraints may involve novel neural architectures, as well as additional domains and tasks. In addition, while we took a step towards measuring connections between EC and representations of natural language, more sophisticated translation mechanisms could uncover deeper relationships. Lastly, given the rich literature that has found IB-efficient human communication in many domains, and our im-proved combinatorial codebook within VQ-VIB, future work might consider connections between EC and different parts of speech.

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

## A  TRAINING HYPERPARAMETERS

Here, we report the hyperparameters used in our experiments. In general, hyperparameters were chosen to maximize self-play performance on the training task, without considering performance on more challenging validation setups (e.g., with more distractors or out-of-distribution images). Thus, we were able to fairly test how successfully learning one task enabled agents to generalize to other settings.

We used a Resnet18, available through TorchVision that had been pretrained on ImageNet, for extracting 512-dimensional features from images. These features were pre-calculated and cached at the start of all training runs. We trained a VAE with a symmetric encoder and decoder with fully connected ReLU layers of dimensions 128, 64, and 32 to reconstruct such features. Thus, in reference games, the speaker and listener actually observed the result of taking an image, passing it

through the Resnet18 feature extractor, and then passing it through the VAE reconstruction to get a slightly noisy version of the true features.

All speakers were parametrized with a common backbone comprising three, 64-dimensional fully-connected ReLU layers, feeding into the communication "head." Details of the particular head architectures are included below. The communication decoder comprised 2 fully-connected ReLU layers with hidden dimension 64, mapping from communication vectors to reconstructed images. The listener agent comprised two separate linear layers that mapped from communication or image features to 16 dimensions; the listener's output was computed via the softmax of the cosine similarity between the communication embedding and each image's embedding.

The decoding loss was calculated as the mean squared error (MSE) between the decoder's output and the speaker's observation. The classification loss was the categorical crossentropy of the listener's output and the onehot label for which candidate image was the target. Agents were trained with batch size 64 until convergence. In the `VG` domain experiments, we trained agents for 20,000 batches; in the `topname` experiments, we trained agents for 100,000 batches.

## A.1  VQ-VIB Speaker

The VQ-VIB Speaker generated communication vectors by first sampling in a 64-dimensional latent space using the reparametrization trick for sampling from a Gaussian and then quantizing to one of the vectors in the codebook. We parametrized VQ-VIB agents with 1024 codebook entries.

Traditional VQ architectures sometimes suffer from "codebook collapse" in which only a small fraction of quantized vectors are used. We found that simply using a small positive $\lambda_C = 0.01$ overcame such issues, likely because regulating complexity increased the stochasticity of encodings, preventing premature settling to local minima.

As in standard VQ architectures, we set the hyperparameter $\beta$ to tradeoff between the committment and embedding losses. In experiments for $\lambda_I > 0$, we set $\beta = 0.25$, the default value. In experiments for $\lambda_I = 0$, we set $\beta = 0.01$; this lower value allowed prototypes to move more, which appeared necessary given the weaker training signal when the informativeness loss was not present.

In all experiments, we used an Adam optimizer with default parameters

## A.2  Onehot Speaker

Although the results in our main paper focused on VQ-VIB agents, we conducted some baseline experiments with onehot-based communication (results reported in Appendix C). The onehot speaker produced onehot vectors by passing through a "hard" Gumbel-softmax layer. We used a 1024-dimensional Gumbel-softmax layer to allow for up to 1024 unique messages.

Training agents with default settings for this layer often failed to achieve better-than-random chance; the speaker often devolved to outputting the same message for all inputs. We explored several methods for addressing such failures, including losses for the message entropy (as discussed in Eccles et al. (2019), among others) and increasing the temperature parameter of the Gumbel softmax layer (as explored in reference games by Chaabouni et al. (2021a), among others). Sweeping over entropy weights from 0.0001 to 0.1 (at powers of 10), and temperatures in the set `[0.1, 1.0, 10.0, 20.0, 50.0]` we achieved the best self-play results by penalizing the message entropy with weight 0.001 and setting the temperature to 20 throughout training, while setting $\lambda_C = 0$. We note that onehot-based communication appeared much more sensitive to hyperparameter tuning than VQ-VIB agents.

Agents were trained using an Adam optimizer with default hyperparameters, except for the learning rate, which we set to 0.0001. This rate led to higher self-play scores compared to 0.001 (the default, which led to unstable training that caused communication collapse) and to 0.00001, which never achieved greater-than-random scores.

## A.3  Prototype Speaker

Similarly to using onehot agents as a baseline, we conducted additional experiments with prototype-based agents, proposed by Tucker et al. (2021). Such agents used an internal Gumbel-softmax layer

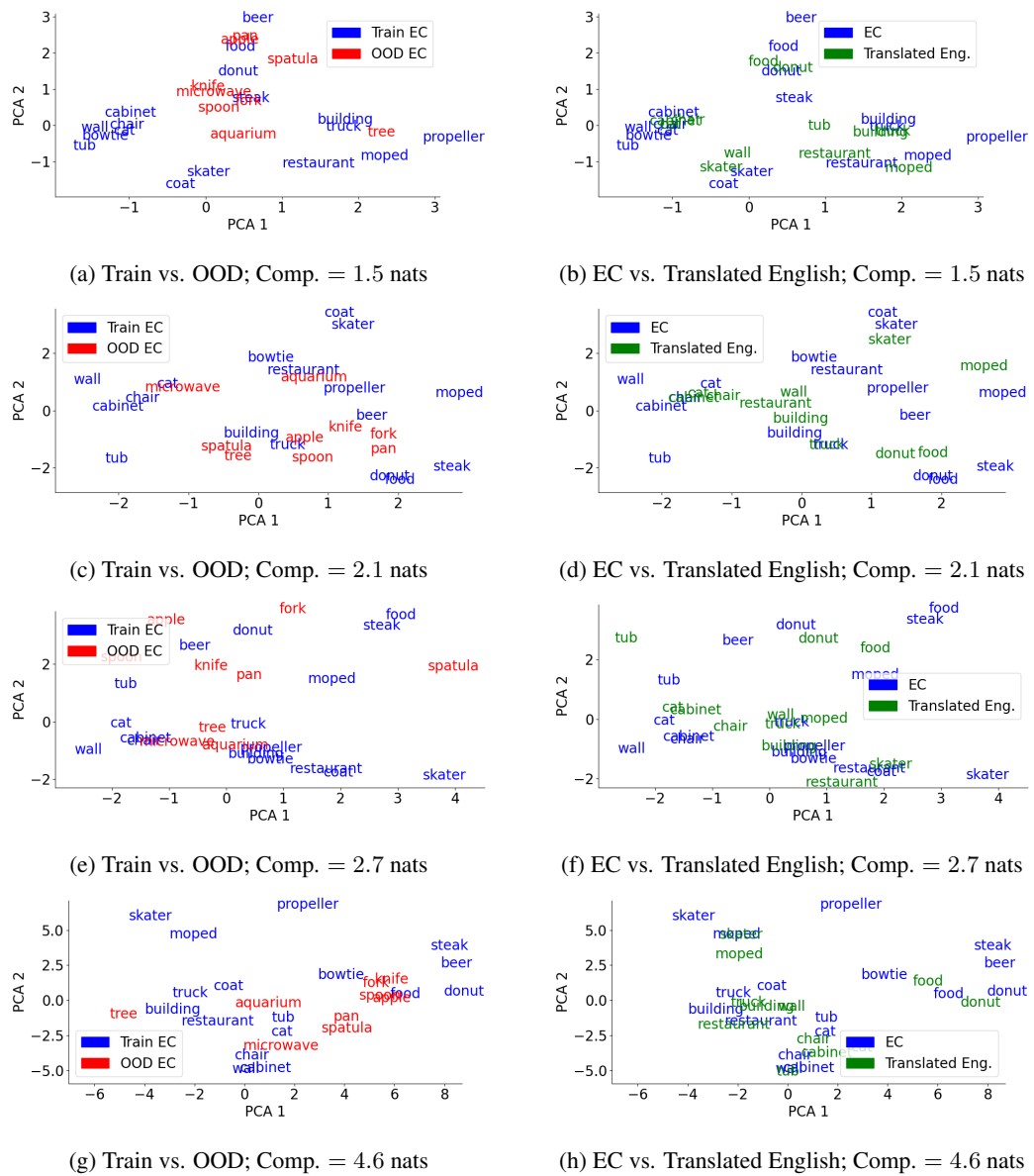

Figure 8: 2D PCA of communication for OOD inputs (left) or translated GloVe embeddings (right) at various complexity levels (different rows).

with 1024 units to select a trainable prototype in 64-dimensional communicative space (to match VQ-VIB). We used default parameters for all settings, except the Adam optimizer learning rate which, as for onehot agents, we set to 0.0001. We also set $\lambda_C = 0$ to encourage highly informative (and complex) communication.

## B  APPENDIX: VISUALIZATION OF LEARNED COMMUNICATION

In this section, we further explored visualization of learned communication via the visualization method used for Figure 6, wherein we recorded the mean EC vector for images associated with different topnames. In Figure 8, we plotted the 2D PCA of EC for in-distribution images (blue), EC for OOD images (in red), and the result of translating English words into the EC space (in green).

Table 2: Team utility in VG reference games, for different speaker types. Onehot and prototype agents converged to lower-complexity and lower-utility communication than VQ-VIB agents.

| Speaker | $\lambda_I$ | Test C 2 | 16 | 32 | Complexity |
|---|---|---|---|---|---|
| Onehot | 0.1 | 0.63 (0.05) | 0.07 (0.01) | 0.05 (0.01) | 0.27 (0.13) |
| | 1.0 | 0.65 (0.04) | 0.14 (0.02) | 0.08 (0.01) | 0.39 (0.11) |
| | 10.0 | 0.87 (0.01) | 0.30 (0.01) | 0.18 (0.01) | 1.27 (0.03) |
| | 100.0 | 0.85 (0.02) | 0.26 (0.02) | 0.15 (0.02) | 1.14 (0.12) |
| Proto. | 0.1 | 0.91 (0.01) | 0.38 (0.03) | 0.23 (0.03) | 1.32 (0.07) |
| | 1.0 | 0.92 (0.00) | 0.43 (0.01) | 0.26 (0.01) | 1.71 (0.04) |
| | 10.0 | 0.93 (0.00) | 0.50 (0.01) | 0.33 (0.02) | 2.01 (0.10) |
| | 100.0 | 0.95 (0.00) | 0.60 (0.02) | 0.42 (0.01) | 2.26 (0.04) |
| VQ-VIB | 0.1 | 0.88 (0.01) | 0.31 (0.01) | 0.17 (0.01) | 1.11 (0.02) |
| | 1.0 | 0.95 (0.00) | 0.59 (0.00) | 0.44 (0.00) | 1.89 (0.02) |
| | 10.0 | **0.99** (0.00) | **0.87** (0.00) | **0.80** (0.00) | 4.51 (0.11) |

Across models of varying complexity, we consistently found that agents, which had been trained on images of walls, dressers, steaks, etc. (but not cribs, spoons, forks, etc.) learned a semantically-meaningful communication space. Words for household items like "dresser" and "'cabinet" are near each other, while foods form another distinct cluster. Furthermore, this semantic space generalized to held-out classes as well. An image of an apple, for example, induced similar communication to images of steaks, donuts, and food in the training set. This explains how agents were able to generalize to held-out classes so well: the form of communication vectors encoded information.

Inspection of the translated GloVe embeddings gives a sense of what EC listeners observed in translation experiments. While admittedly imperfect, the linear translator model was largely able to locate translated communication in the right semantic area. For example, in Figure 8 d, the translated EC for "donut" and "food" are not perfectly aligned with EC for donuts or food, but the two representations are still near each other.

## C  APPENDIX: OTHER COMMUNICATION ARCHITECTURES

Although the main focus of our work is on the relationship between information-theoretic properties of EC and generalization or translation (and therefore independent of the exact form of communication), we performed some experiments with onehot- and prototype-based (the architecture introduced by Tucker et al. (2021)) communication. Results with such agents reinforced the main findings of our paper establishing a link between complexity and generalization, regardless of the specific speaker architecture. Further, our results indicated that VQ-VIB agents enabled far greater control over communication complexity compared to onehot or prototype agents.

We replicated the VG experiments from Section 6.1. Teams were trained with $C = 2$, with candidates drawn from distinct VG categories, and tested with varying $C$ for candidates with no distinction enforced. Results are included in Table 2, including results for VQ-VIB (for $n = 4$).

The most important result in Table 2 is that the trend between greater complexity and greater utility holds for all agent architectures. That is the primary focus of this work, and this table indicates that specific neural architectures do not affect this finding. Beyond that primary result, secondary effects from different speaker types are visible. First, for the same $\lambda_I$, onehot agents tended to converge to lower-complexity and lower-utility communication than VQ-VIB agents. This corroborates findings from Tucker et al. (2022) that the VQ-VIB architecture appears to have a natural inductive bias towards more complex communication. Second, onehot agents seemed to require more complex communication for the same utility as VQ-VIB agents. For example, onehot communication for $\lambda_I = 10$ achieved comparable utility to VQ-VIB for $\lambda_I = 0.1$, but at 1.27 nats instead of 1.11 nats for VQ-VIB. Third, even for very high $\lambda_I$, we were never able to induce as complex or high utility communication for onehot agents as VQ-VIB agents, and in fact high $\lambda_I$ appeared to worsen performance. Lastly, prototype agents to a large extent reproduced the trends present in VQ-VIB agents, although still at lower complexities. Given their underlying similarities of using discrete representations in a continuous space, this result is unsurprising.

Table 3: Team utility for OOD for `topname` agents, for different speaker types and selected $\lambda_I$. Across architectures, increasing $\lambda_I$ led to more complex communication that generalized better, but VQ-VIB (for $n = 4$) achieved the best performance.

| | | **Test C** | | | |
|---|---|---|---|---|---|
| **Speaker** | $\lambda_I$ | 2 | 16 | 32 | **Complexity (nats)** |
| Onehot | 0.1 | 0.53 (0.05) | 0.07 (0.00) | 0.03 (0.00) | 0.19 (0.25) |
| | 1.0 | 0.71 (0.08) | 0.17 (0.05) | 0.10 (0.04) | 0.92 (0.35) |
| | 10.0 | 0.78 (0.02) | 0.25 (0.03) | 0.16 (0.00) | 1.30 (0.12) |
| | 100.0 | 0.77 (0.01) | 0.24 (0.00) | 0.15 (0.00) | 1.26 (0.15) |
| Proto. | 0.1 | 0.86 (0.03) | 0.36 (0.03) | 0.24 (0.04) | 1.19 (0.08) |
| | 1.0 | 0.89 (0.00) | 0.44 (0.02) | 0.29 (0.01) | 1.53 (0.14) |
| | 10.0 | 0.92 (0.00) | 0.53 (0.05) | 0.40 (0.05) | 1.91 (0.05) |
| | 100.0 | 0.93 (0.02) | 0.58 (0.07) | 0.43 (0.06) | 2.50 (0.06) |
| VQ-VIB | 0.1 | 0.82 (0.23) | 0.29 (0.03) | 0.16 (0.01) | 1.55 (0.23) |
| | 1.0 | 0.91 (0.02) | 0.51 (0.06) | 0.37 (0.04) | 2.73 (0.06) |
| | 10.0 | 0.97 (0.00) | 0.80 (0.06) | 0.71 (0.08) | 4.46 (0.51) |
| | 100.0 | **0.98** (0.01) | **0.80** (0.03) | **0.72** (0.04) | 4.46 (0.24) |

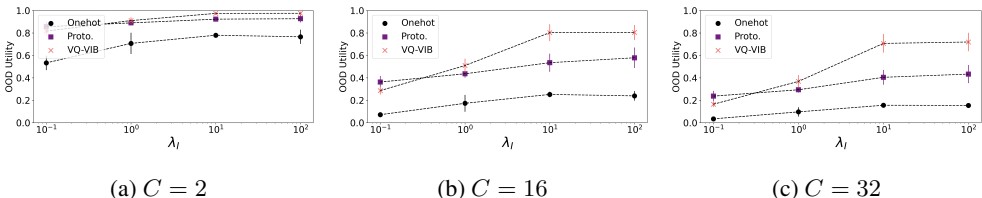

(a) $C = 2$          (b) $C = 16$          (c) $C = 32$

Figure 9: Out-of-distribution utility as a function of $\lambda_I$ (plots for different $C$). For all architectures and $C$, increasing $\lambda_I$ first increased utility before starting to plateau.

Similar trends appeared in `topname` experiments with onehot- and prototype-based communication using the same 20% generalization evaluation procedure. First, for various $\lambda_I$, we found that VQ-VIB converged to greater complexity and utility than other architectures (Table 3, and depicted graphically in Figure 9). Second, as in the `VG` experiments, we found that all agent architectures achieved roughly similar utility for the same complexity, but that non-VQ-VIB agents failed to learn as complex communication (Figure 10). (Interestingly, prototype agents seemed to learn more complex communication than onehot, but not as complex as VQ-VIB.) Note that simply increasing $\lambda_I$ failed to produce the desired effect for onehot agents, which converged to similar behavior for $\lambda_I = 10$ and $\lambda_I = 100$. Jointly, these results indicate that tuning $\lambda_I$ was an effective mechanism for increasing the complexity and informativeness of communication, and that VQ-VIB was best able to learn the highest-utility communication.

Lastly, translation performance with onehot-based agents was predictably poor, with utility for $C = 2$ never exceeding 75%, even for $\lambda_I = 100$. Such failure can be attributed to two limitations of onehot agents. First, as already shown in earlier experiments, onehot agents learned lower-complexity communication than VQ-VIB agents, which limited the maximum utility. Second, specifically in the context of translation, communicating via onehot vectors inherently limits the semantic relationships between messages. VQ-VIB agents, with discrete representations in a continuous space, could leverage the alignment dataset to learn a transformation of the whole space. Conversely, because every onehot vector is by definition orthogonal to every other onehot vector, translation necessarily failed to capture relationships between messages.

Ultimately, based on these results, we omitted discussion of onehot- and prototype-based communication from the main paper. Our primary focus was the effect of different complexity and informativeness on generalization; given the consistent results across architectures but more limited range of complexities for non-VQ-VIB agents, VQ-VIB agents simply presented more interesting results.

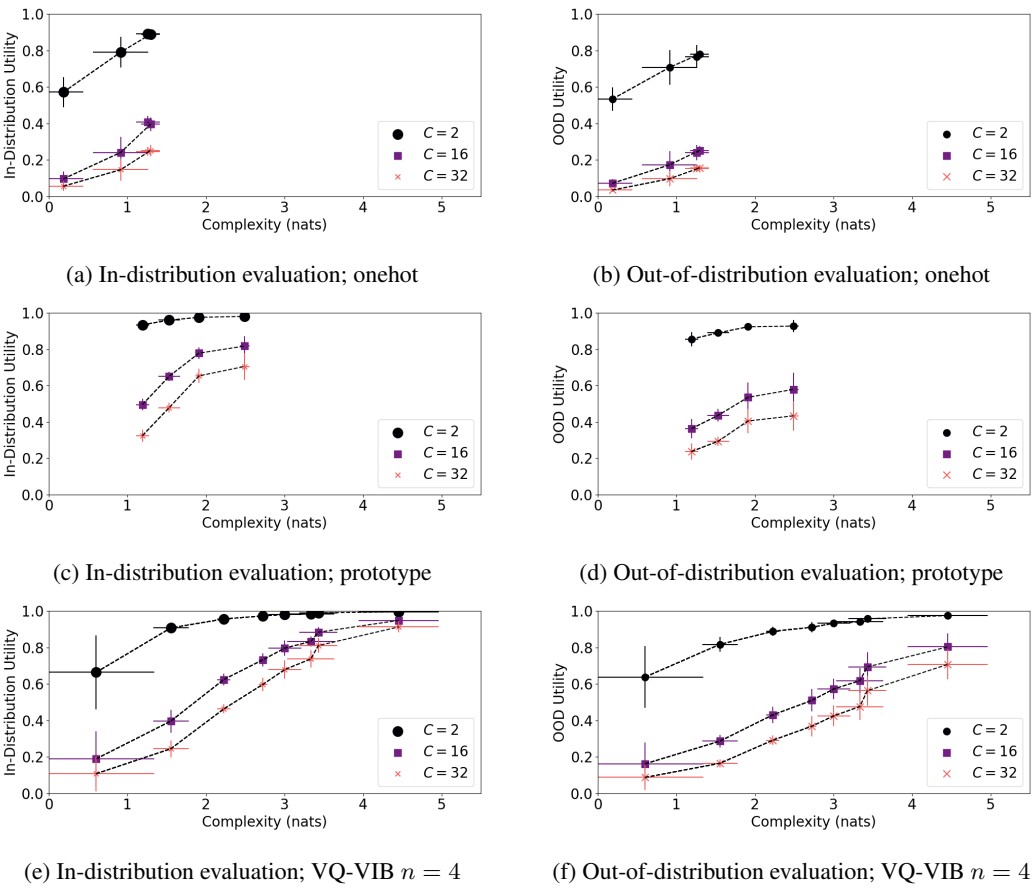

(a) In-distribution evaluation; onehot

(b) Out-of-distribution evaluation; onehot

(c) In-distribution evaluation; prototype

(d) Out-of-distribution evaluation; prototype

(e) In-distribution evaluation; VQ-VIB $n = 4$

(f) Out-of-distribution evaluation; VQ-VIB $n = 4$

Figure 10: In- and out-of-distribution utility for onehot (top), proto (middle), or VQ-VIB agents (bottom). All agents exhibited similar utility for the same complexity, but VQ-VIB agents learned far more complex communication, which allowed them to perform better in harder tasks.

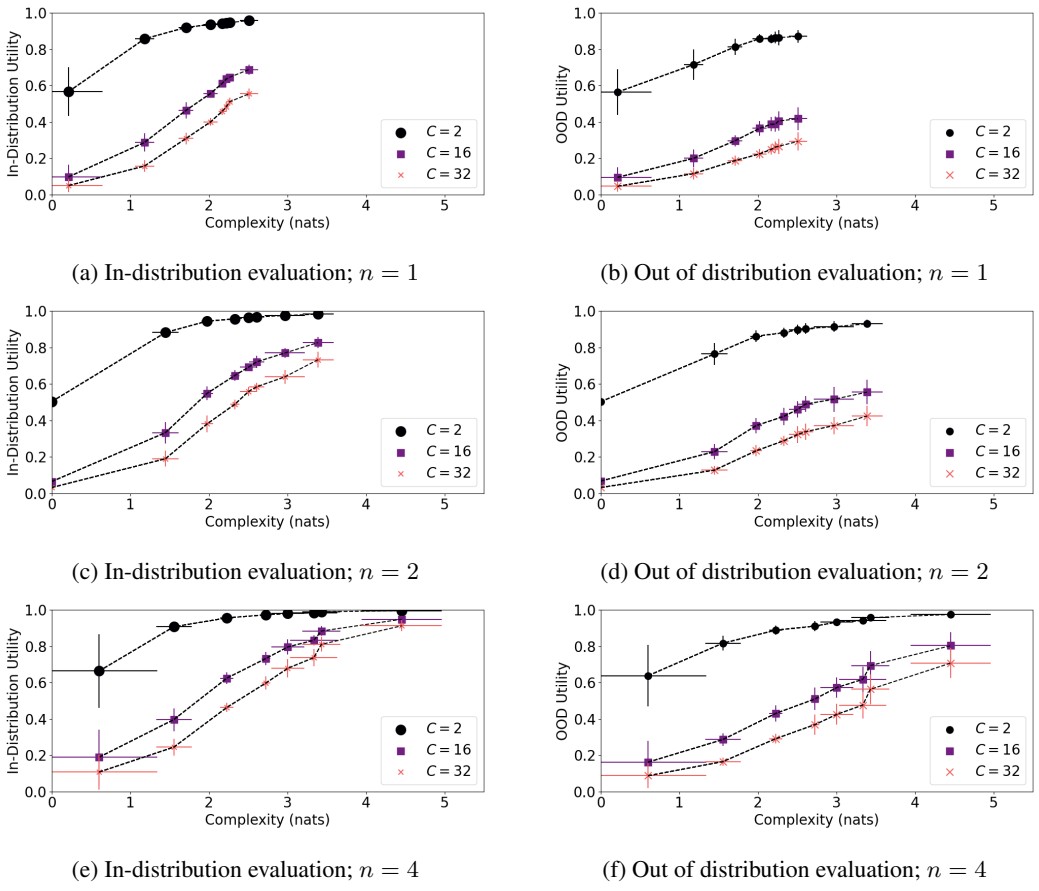

(a) In-distribution evaluation; $n = 1$

(b) Out of distribution evaluation; $n = 1$

(c) In-distribution evaluation; $n = 2$

(d) Out of distribution evaluation; $n = 2$

(e) In-distribution evaluation; $n = 4$

(f) Out of distribution evaluation; $n = 4$

Figure 11: Team performance on in-distribution (a) and out-of-distribution (b) images, for various $n$ (different rows). The general complexity-utility tradeoff remained the same, but using greater $n$ allowed greater complexity.

## D    APPENDIX: COMBINING QUANTIZED VECTORS

In the main paper, we primarily presented results generated for VQ-VIB agents with $n = 4$. In Figure 11, we include results from our `topname` experiments, generated for $n = 1, 2$, and 4. In general, we found that using greater $n$ increased communication informativeness and complexity, which in turn supported greater utility. Most importantly, across $n$, we found a similar relationship between utility and complexity. Thus, we view this architectural change of using larger $n$ as allowing greater control over the complexity and informativeness of communication (which, as we discuss in the main paper, is important for downstream tasks).

## E    APPENDIX: VARYING TRANSLATION ALIGNMENT DATA

In Section 6.3, we presented results for a simulated English speaker, with communication translated for an EC listener. The translator was a linear transformation, fitted with $N$ randomly-drawn examples from the training dataset. In the main paper, we presented results for $N = 100$; here, in Figure 12, we present results for various $N$.

Ultimately, while increasing $N$ helped translation somewhat, the general trends from the main paper held true. That is, as EC complexity increased from 0 to just over 2 nats, team utility increased, but at greater complexity values, team performance was bottlenecked by the English speaker. Furthermore, beyond a certain value, increasing $N$ further did not appear to improve performance, as demonstrated by the nearly identical performance for $N = 100$ and $N = 1000$.

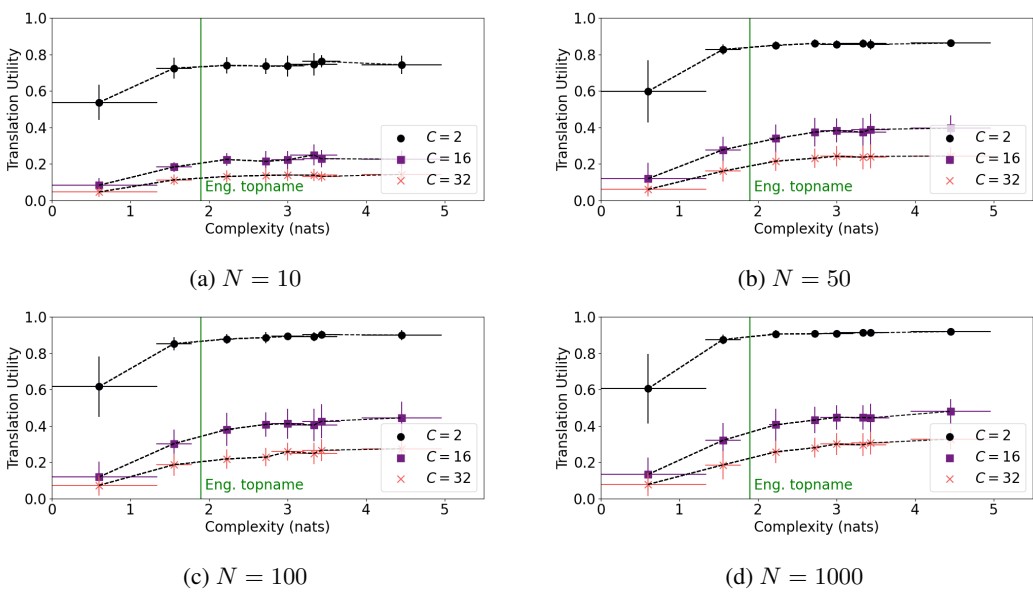

(a) $N = 10$

(b) $N = 50$

(c) $N = 100$

(d) $N = 1000$

Figure 12: Utility of human-agent teams using English GloVe to EC translation, for various translation dataset sizes. Increasing the dataset size helped up to a point, but performance plateaued around 2 nats in all cases.

