# OpenReview forum: "Information-Theoretic Underpinnings of Generalization and Translation in Emergent Communication"
_ICLR.cc/2023/Conference — Submitted to ICLR 2023_

### Official Review · Reviewer_YuEu · 2022-10-25

**Confidence:** 3
**Correctness:** 3
**Technical Novelty And Significance:** 2
**Empirical Novelty And Significance:** 3
**Recommendation:** 6

**Clarity, Quality, Novelty And Reproducibility:**

Overall the paper is well-written and the experiments included are well-motivated and explained. The novelty (in terms of the new proposed method for training EC agents) is somewhat limited, but works well in-practice and the analysis are interesting. To cement the importance of this work, I would only like to see this method and analysis applied to other settings/referential games.

**Strength And Weaknesses:**

- The paper is well-written and has a nice narrative. The idea of allowing multiple vectors is quite simple but works well in pratice, and the overall analysis of how informativeness and complexity in the channel affects performance is insightful.
- The experiments with (simulated) human-agent are really interesting, and are nice approach to interpreting the "abstract" protocols these models learn.

My main criticism of this work, given the simplicity of the approach (that mostly just builds on top of the previously proposed VQ-VIB) I would like to see if it generalizes to multiple benchmarks and settings that emergent communication could be applied to. The authors only study one task/referential game for images, so it’s unclear if the findings surrounding the importance of channel complexity / applicability of their proposed approach generalize to more settings.

**Summary Of The Paper:**

This paper studies how the informativeness and complexity of the emergent communication protocol in two-agent referential games affects the generalization performance and how these messages align with natural language.

In particular, they use the recently proposed framework of VQ-VIB, a variational inference approach using a discrete set of vectors (vector-quantitized) as possible messages for the agents. They extend this framework to allow passing $N$ vectors rather than just one as messages, and show that this allows the communication protocol to have more complexity.  They show that the \lambda parameter in the VQ-VIB objective (associated with the **********informativeness********** of the message) also leads to the communication channel being more complex. Overall, the results suggest that greater complexity in protocol (with either multiple code-vectors or higher \lambda) leads agents solving the referential game with higher performance.

To analyze the impact of the communication channel on out-of-domain generalization, they train agents for the same referential but using, at test time, a higher number of candidate images than trained for and/or candidates belonging to the same category of images. Overall they find that higher protocol complexity leads to agents that generalize better.

Finally they test how aligned this emergent communication are with natural language by building a simple model that “translates” english words to one of the quantitized vectors and passing the associated image category word through this model to get a code and having the receiver agent predict based on this. They find that overall that this simple translation protocol works and that higher complexity in the channel helps but that, unlike with pure vector-based protocols, improvements with complexity plateu earlier.

**Summary Of The Review:**

This paper proposes an extension of VQ-VIB to allow for more complex communication protocols and show that complexity is intrinsically correlated with how well the agents will do at the game. The analysis and experiments are convincing for the single task explored in the paper, but I would like to see this methodology applied more extensively to other tasks and settings.

---

> ### Author Response · Authors · 2022-11-15
> **Thanks; clarification on benchmarks**
>
> We thank the reviewer for these thoughtful and helpful comments, and overall encouraging review! As far as we can tell, the reviewer is overall positive about our work and only has one major concern. Before we address that concern, we feel that it may be useful to clarify that we evaluated agent generalization not only by varying the number of test-time candidates (as noted by the reviewer) but also by holding out specific classes of images during training, and testing on them (see Section 6.2 for results). For example, some agents were trained with images of “beer”, “steak,” and “donut” but not of “knife”, “fork,” or “pan,” upon which they were tested (see Figure 6).
>
> > “My main criticism of this work, given the simplicity of the approach (that mostly just builds on top of the previously proposed VQ-VIB) I would like to see if it generalizes to multiple benchmarks and settings that emergent communication could be applied to. The authors only study one task/referential game for images, so it’s unclear if the findings surrounding the importance of channel complexity / applicability of their proposed approach generalize to more settings.”
>
> We agree that extensions to additional tasks and benchmarks is an important direction for further research, and we have also noted that in our initial submission (Section 7). At the same time, we would like to emphasize that we have not considered only one benchmark but rather three: (1) testing with more distractors; (2) testing on out-of-distribution inputs; and (3) testing the interaction between EC agents and a simulated English speaker. (1) was inspired by generalization to harder tasks, as introduced by [Chaabouni et al. 2021](https://openreview.net/pdf?id=AUGBfDIV9rL) and (2) measures important generalization capabilities that is similar to the cross-dataset generalization evaluated in  [Chaabouni et al. 2021](https://openreview.net/pdf?id=AUGBfDIV9rL), but distinct due to the intentionally semantically-distinct nature of our training and testing sets . In addition, (3) is a qualitatively different benchmark, especially when considering the unique characteristics of the ManyNames dataset that led us to pick it for our experiments. In contrast to standard image classification datasets (e.g., CIFAR), which consist of very coarse and artificially curated labels, the ManyMames dataset consists of large-scale free-naming data. These data were collected using an experimental paradigm from cognitive science, in which participants are free to generate any word they wish to describe the input image. Therefore, the data reflects the rich semantic structure organically represented by native English speakers, and handling this semantic structure during test time poses a new kind of challenge for EC agents.
>
> As explained in Section 5, our decision to focus specifically on the ManyNames dataset is motivated by the fact that this is the only dataset we are currently aware of that consists free-naming data with respect to a large amount of images, spanning multiple different domains (e.g., plants, people, furniture, etc.). We are currently working on extending our results to navigation tasks inspired by  [Tucker et al. 2022](https://openreview.net/pdf?id=O5arhQvBdH). While we are confident that our approach will generalize to those settings, that requires a substantial amount of additional work, including collecting new free-naming human data, and we therefore believe that it could not fit within the scope of a single ICLR paper.
>
> Finally, we wish to highlight that VQ-VIB is a very recent method that has not yet been fully explored in the literature. We believe that our results provide a substantial extension of the initial experiments performed by [Tucker et al. 2022](https://openreview.net/pdf?id=O5arhQvBdH) (who first introduced VQ-VIB), and show several important properties of VQ-VIB that were unknown before. In addition, we believe that our new benchmarks could benefit the EC community more generally, as they can be applied to many EC methods (as we already started to do in Appendix C).

---

### Official Review · Reviewer_QWC4 · 2022-10-25

**Confidence:** 4
**Correctness:** 4
**Technical Novelty And Significance:** 2
**Empirical Novelty And Significance:** 2
**Recommendation:** 3

**Clarity, Quality, Novelty And Reproducibility:**

### Clarity
- Although writing itself is clear, the organization of high-level concepts is a little difficult to follow.
  Presenting a clear roadmap would help fix this.
  - Between sections 2 and 3, a little over two pages are spent revisiting prior work -- this can be compressed.
- The visualizations in the work are helpful and aid in understanding the results.
  - Figure 1 is especially informative
- The contributions and hypotheses are clear at a basic level, but they are not explained in enough detail to make clear what the criteria for success are.
- The results seemed to be making a lot of distinct but related claims in rapid succession.
  As a result, I had difficulty incorporating the result into the big picture of the paper.
### Quality
- The empirical result generally corroborate what the hypotheses and intuitions would suggest.
  - Nevertheless, since there are not clear criteria for positive results (since the hypotheses are not well-defined), it is not easy to determine to what degree the empirical data justifies the claims.
  - The number and nature of the experiments has the wrong balance between quantity and quality: many different experiments off smaller bits of evidence for the claims rather than many related experiments supporting one big claim.
  - More quantitative analysis of the results would make them more convincing.
- The signalling game is a relatively simple/easy environment which limits the robustness of the claims.
  Would we the see the same relationships in more complex environments?
- The multi-component message extension to VQ-VIB and the translation method are simple and effective.
### Novelty
- This work furthers understanding of a newer framework within EC, namely VQ-VIB.
- The topic and approach are not particularly innovative.
  Papers of the form "we explore the relationship between X and Y" have characterized emergent language research for years, and work needs to either go beyond that basic recipe or explain why it is important to know the relationship between X and Y and to do so thoroughly.
### Reproducibility
- Looked at (did not run) repository with code.
  `README` present with most information necessary for running the code.
  Only notable item missing with a specification of the environment used (e.g., `requirements.txt`, `environment.yml`).


**Strength And Weaknesses:**

### Strengths
- [major] Empirical evaluation generally supports the hypotheses.
- [minor] The extensions to VQ-VIB (translation and multi-component messages) are simple and appear to be effective.
### Weaknesses
- [major] The paper makes too many claims which means that each claim receives only a small amount of empirical evaluation, analysis, and discussion.
  - I think the primary direction forward would be to take one of the claims, and take the whole paper to address it thoroughly.
    Give a robust explanation of why that claim is important and how important future work will be enabled by the contributions of the paper (i.e., do not just say "we could extend this with X, Y, and Z").
  - Example: "increasing informativeness ... allows EC agents to better generalize to novel settings and more challenging tasks" -- this is an interesting claim that would require a whole battery of experiments to prove convincingly.
    Increasing the number of distractors in a signalling game does not convincingly count as a novel setting or more challenging task.
    Instead, it would be best to see multi-step, multi-agent environments with different objectives that are not just a simple extension of the training environment.
  - Example: "increasing EC informativeness only improves team performance up to a certain threshold, corresponding to English informativeness-complexity tradeoff" -- again, very interesting, but one would need to provide a robust characterization of what the trade-off is in English in the first place.
    Furthermore, how do we evaluate this tradeoff beyond a specific environment and dataset; it would be good to first show this on synthetic data or in a theoretical setting before validating that it works the same way on real data.
  - Example: "the structure of emergent VQ-VIB communication vectors encodes some semantic similarities and facilitates open-domain communication, similar to word embeddings in natural language" -- a very worthwhile direction that only receives a diagram and a paragraph rather than thorough treatment analyzing the structure of embedding spaces from multiple directions.


**Summary Of The Paper:**

This work explores the relationship between emergent communication informativeness and complexity, hyperparameters of the vector-quantized variational information bottleneck (VA-VIB) method of training EC agents, and downstream metrics like generalizability and translatability with human language.
It finds that as the informativeness and complexity of emergent communication increases, it correlates with increased generalizability and translatability.
In support of these experiments, the paper extends VQ-VIB to multi-component messages and introduces a method for translating between embedding spaces of different languages (viz., English and an emergent language).


**Summary Of The Review:**

This paper shows a handful of positive empirical results in support of the intuitions presented about hyperparameters in the VQ-VIB framework.
Nevertheless, the treatment of each contribution is relatively brief which does not communicate a sufficiently robust understanding of what is going or impress the importance of the contribution to emergent communication research as a whole.
As a result, the paper falls into a common category of emergent communication paper which presents an array of experiments and claims which, while interesting, are unlikely to be impactful.


### Misc. Notes
- Please use $\mathbb{R}$ (`\mathbb{R}`) for the real numbers

---

> ### Author Response · Authors · 2022-11-17
> **Response part 1:**
>
> We thank the reviewer for these thoughtful comments, and appreciate the positive feedback as well as the concerns and suggestions for extending the research. Below are our responses to each concern raised by the reviewer.
>
> > “The paper makes too many claims which means that each claim receives only a small amount of empirical evaluation, analysis, and discussion.
> I think the primary direction forward would be to take one of the claims, and take the whole paper to address it thoroughly. Give a robust explanation of why that claim is important and how important future work will be enabled by the contributions of the paper (i.e., do not just say "we could extend this with X, Y, and Z").”
>
> We understand the reviewer’s suggestions on the tension between broader vs. deeper research, and agree that many of the ideas we explore in this work are rich enough that they could warrant whole papers on their own. Ultimately, we chose to write a paper about information-theoretic underpinnings of EC, rather than focus exclusively on a specific feature of EC like generalizability alone or translatability alone. Our central thesis is that controlling the informativeness-complexity tradeoff of EC systems is important for several characteristics of EC. Our generalization and translation experiments, therefore, are not disconnected tasks but rather evidence of a common theme. If we had only explored generalization, for example, this paper might make stronger claims about generalization specifically but fail to indicate the more important and general connection between information-theoretic principles and EC performance. We have updated the abstract in our latest revision to better highlight this framing.
>
> In addition, we wish to highlight that we have provided additional experiments in the appendices, including an extensive comparison with baselines (Appendix C), which was specifically noted by reviewer cAgg as convincing.To better highlight these comparisons, we have revised the paper to include some of these results in Section 6 of the main paper (Figure 5c) and we now point more clearly to full results in the appendices.
>
> > “Example: "increasing informativeness ... allows EC agents to better generalize to novel settings and more challenging tasks" -- this is an interesting claim that would require a whole battery of experiments to prove convincingly. Increasing the number of distractors in a signalling game does not convincingly count as a novel setting or more challenging task. Instead, it would be best to see multi-step, multi-agent environments with different objectives that are not just a simple extension of the training environment.”
>
> While we agree that extending our work to multi-step and multi-agent environments with different objectives is one way to consider more challenging settings and tasks, and this is definitely an exciting direction for future research, there are also many other ways of doing that and it is unclear if one could explore all these possibilities within the scope of a single paper. We therefore do not claim that we show better generalization to every possible extension to a novel setting and more challenging task, but rather to a few extensions that have been identified in the literature as challenging. For example, varying the number of distractors was inspired by prior work by [Chaabouni et al. 2021](https://openreview.net/pdf?id=AUGBfDIV9rL), who explicitly describe such testing as ``scaling up task difficulty.’’ Literature in contrastive learning similarly explores using more distractors during training as a harder task designed to induce richer representations [He et al. 2019](https://ieeexplore.ieee.org/document/9157636). In addition to changing task difficulty by varying the number of candidates, we also tested agents on new types of objects from domains that were never seen by the agents during training. Testing models on such out-of-distribution inputs is commonly considered as a harder generalization task (see e.g., [Chaabouni et al. 2021](https://openreview.net/pdf?id=AUGBfDIV9rL)], who trained and evaluated agents on different image datasets, and [Bouchacourt and Baroni](https://aclanthology.org/D18-1119.pdf) who hold out subclasses of ImageNet for evaluation). Given the current state of the field and the fact that very recent papers on EC have considered our generalization tasks challenging, we believe that our experiments provide an important and timely contribution that would help the community advance EC even further, as the reviewer envisions for future research.

---

> > ### Author Response · Authors · 2022-11-17
> > **Response part 2:**
> >
> > > “Example: "increasing EC informativeness only improves team performance up to a certain threshold, corresponding to English informativeness-complexity tradeoff" -- again, very interesting, but one would need to provide a robust characterization of what the trade-off is in English in the first place. Furthermore, how do we evaluate this tradeoff beyond a specific environment and dataset; it would be good to first show this on synthetic data or in a theoretical setting before validating that it works the same way on real data.”
> >
> > As we explain in the paper, our work builds on extensive prior work that has carefully studied the informativeness-complexity tradeoff of human naming systems across many languages and  in a variety of semantic domains, including names for objects  (e.g., Zaslavsky et al., 2018 and 2019, and see more references in the paper).  How to evaluate the tradeoff for English more broadly, beyond a specific domain, is an open question on its own and it is beyond the scope of this work. For our particular domain, we estimated the complexity and informativeness using standard methods, cited in the paper.
> >
> > > “Example: "the structure of emergent VQ-VIB communication vectors encodes some semantic similarities and facilitates open-domain communication, similar to word embeddings in natural language" -- a very worthwhile direction that only receives a diagram and a paragraph rather than thorough treatment analyzing the structure of embedding spaces from multiple directions.”
> >
> > First, we would like to highlight that we have included further evaluation of the embedding space beyond the diagram and paragraph in the main text (see Appendix B). This supplementary analysis corroborates the trends presented in the main text, at different complexity levels, and further shows comparisons to translated word embeddings. Second, we note that the translation experiment results also provide useful insight as to the semantically-meaningful structure of the EC space. Given that the translator model is just a linear regression, the fact that team performance is so high when using this translation implies that there is at least some semantic structure in the EC space. If all VQ-VIB vectors were orthogonal (as in onehot communication, which encodes no relationships between vectors), linear translation would fail. Thus, our translation experiments are useful not only for their main purpose of showing English-EC alignment, but also for revealing similarities between word embeddings and EC vectors.
> >
> > > "Although writing itself is clear, the organization of high-level concepts is a little difficult to follow. Presenting a clear roadmap would help fix this."
> >
> > We would greatly appreciate it if the reviewer could be more specific as to which high-level concepts are not well organized. As for the roadmap, we could add a paragraph explaining the structure of the paper, but we’re not sure if that is really necessary given that our paper follows one of the more common structures in machine learning papers (introduction, related work, background, technical approach, experimental design, results, conclusions). We would be happy to adjust the final version based on the reviewer’s response, and believe this could be addressed with a very minor revision.
> > Between sections 2 and 3, a little over two pages are spent revisiting prior work -- this can be compressed.
> > Much of the background that is crucial for understanding our work is not commonly known in the ICLR community. For example, VQ-VIB was first published only very recently, and some of the other background is based on broader interdisciplinary literature. Therefore, we feel that it is important to keep a comprehensive background section to make our work more self-contained and widely accessible to the ICLR community.
> >
> >
> > > "The visualizations in the work are helpful and aid in understanding the results. Figure 1 is especially informative"
> >
> > Thank you! We are very happy to hear this.
> >
> > > "The contributions and hypotheses are clear at a basic level, but they are not explained in enough detail to make clear what the criteria for success are."
> >
> > In all cases our measure of success is team performance during test time, where the team could either be two pre-trained agents or a pre-trained listener agent and a simulated English speaker. We believe this evaluation is standard in EC literature, but we have also clarified this point in the current version.
> >
> > > "The results seemed to be making a lot of distinct but related claims in rapid succession. As a result, I had difficulty incorporating the result into the big picture of the paper."
> >
> > We hope that our adjusted abstract and conclusion section, as well as our summary of contribution in the introduction, help address this point.

---

> > > ### Author Response · Authors · 2022-11-17
> > > **Response part 3:**
> > >
> > > > "The empirical result generally corroborate what the hypotheses and intuitions would suggest.
> > > Nevertheless, since there are not clear criteria for positive results (since the hypotheses are not well-defined), it is not easy to determine to what degree the empirical data justifies the claims."
> > >
> > > Our framing intends to convey the high-level intuition that motivated our experiments rather than formal hypotheses. We agree that the term “hypothesis” is used a bit loosely, and we will try to think of a better term for the final version. Having said that, the evaluation criteria is well-defined (team performance, as clarified above) and we believe that the conclusions are supported by our quantitative empirical results (especially when considering the comparison with baselines).
> > >
> > > > "The number and nature of the experiments has the wrong balance between quantity and quality: many different experiments off smaller bits of evidence for the claims rather than many related experiments supporting one big claim.
> > > More quantitative analysis of the results would make them more convincing."
> > >
> > > We feel this comment mirrors the earlier comment raised by the reviewer on focusing on one idea or many. We hope that our responses above have addressed this point to the reviewer’s satisfaction.
> > >
> > > > "The signalling game is a relatively simple/easy environment which limits the robustness of the claims. Would we the see the same relationships in more complex environments?"
> > >
> > > We are indeed excited about this direction for future research (as we have also noted in Section 7), and are currently exploring other types of environments or games.
> > >
> > > > "The multi-component message extension to VQ-VIB and the translation method are simple and effective."
> > >
> > > Thank you!
> > >
> > > > "The topic and approach are not particularly innovative. Papers of the form "we explore the relationship between X and Y" have characterized emergent language research for years, and work needs to either go beyond that basic recipe or explain why it is important to know the relationship between X and Y and to do so thoroughly."
> > >
> > > We feel that reducing our work to  “explor[ing] the relationship between X and Y”  is an abstraction that neglects some of the key contributions and insights of our work. Specifically, our work aims to understand how general information theoretic principles, which are believed to guide human language evolution, may help advance EC and guide it toward more human-like communication. This could have broader implications to several fields.
> > >
> > > For example, understanding how EC agents respond to novel inputs is an important problem that many EC researchers have considered ([Chaabouni et al. 2021](https://openreview.net/pdf?id=AUGBfDIV9rL), [Bouchacourt and Baroni 2018](https://aclanthology.org/D18-1119.pdf), [Tucker et al. 2021](https://openreview.net/pdf?id=hsqZ5v8PFyQ)), and recent research explores the connections between EC and NLP ([Tucker et al. 2021](https://openreview.net/pdf?id=hsqZ5v8PFyQ), [Yao et al. 2022](https://openreview.net/pdf?id=49A1Y6tRhaq)). Our work shows that incorporating informational constraints in EC may help agents generalize to novel inputs, and it suggests a new path for further connecting EC with NLP via semantic embedding spaces.
> > >
> > > In addition, research in human naming systems indicates the importance of information-theoretic constraints in human language ([Zaslavsky et al. 2018](https://www.pnas.org/doi/10.1073/pnas.1800521115), [Zaslavsky et al. 2022](https://academic.oup.com/jole/advance-article/doi/10.1093/jole/lzac001/6566271), [Zaslavsky et al. 2019](https://arxiv.org/abs/1905.04562), [Mollica et al. 2021](https://www.pnas.org/doi/10.1073/pnas.2025993118), and others cited in our main paper). Hence, our work, seeking to connect broadly-desired properties of EC to important principles from cognitive science, both addresses concrete problems within the EC field and more generally provides interdisciplinary insight into pressures on communication.
> > >
> > > > "Looked at (did not run) repository with code. README present with most information necessary for running the code. Only notable item missing with a specification of the environment used (e.g., requirements.txt, environment.yml)."
> > >
> > > We thank the reviewer for their careful review of  our code. We will adjust the code for easy installation upon acceptance.

---

> > > > ### Author Response · Authors · 2022-11-17
> > > > **Response part 4:**
> > > >
> > > > > "Nevertheless, the treatment of each contribution is relatively brief which does not communicate a sufficiently robust understanding of what is going or impress the importance of the contribution to emergent communication research as a whole. As a result, the paper falls into a common category of emergent communication paper which presents an array of experiments and claims which, while interesting, are unlikely to be impactful."
> > > >
> > > > We hope that our earlier clarifications regarding the goals of this work and the extensive results that appear in the appendices have addressed the reviewer’s concerns. Predicting a paper’s impact is known to be an extremely hard problem, however we believe that our work would make a valuable contribution to the ICLR community and we hope to have the opportunity to share it with the community.
> > > >
> > > > > "Please use $\mathbb{R}$ (\mathbb{R}) for the real numbers"
> > > >
> > > > Thank you for pointing this out. We have updated the notation in the latest revision.

---

> > ### Comment · Reviewer_QWC4 · 2022-11-17
> > **Need for focus in claims, experiments, and analysis**
> >
> > > Ultimately, we chose to write a paper about information-theoretic
> > > underpinnings of EC, rather than focus exclusively on a specific feature of
> > > EC like generalizability alone or translatability alone. Our central thesis
> > > is that controlling the informativeness-complexity tradeoff of EC systems is
> > > important for several characteristics of EC.
> >
> > This is still, in my estimation, a weakness in the approach of the paper.
> > Emergent language is a difficult field to research due to its open-ended and
> > highly variable nature.  I think this warrants making small but
> > well-investigated claims through careful experimentation.  A paper which tries
> > to broadly connect experiments with a common theme does not satisfy this.
> >
> > > We would greatly appreciate it if the reviewer could be more specific as to
> > > which high-level concepts are not well organized.
> >
> > More specifically, after reading the abstract and the introduction of the paper
> > (reread both after the revisions, as well), I do not have a clear vision of
> > what the paper is going to be doing (beyond the general intro, related work,
> > methods, results, discussion).  For example, take
> > > [W]e hypothesize that taking into account informativeness could improve EC
> > > generalizability to novel settings[.]
> >
> > After reading this, I am not sure what "taking into account informativeness"
> > entails, how we might measure generalizability, or what the novel settings are.
> > I would expect to see a level of detail more along the following lines,
> > > VQ-VIB controls the relative informativeness pressure of communication via
> > > a real-valued hyperparameter.  We hypothesize that there is a positive
> > > correlation between the informativeness hyperparameter and generalizability
> > > (i.e., performance in novel settings).  Specifically, we use a signalling
> > > game with a higher number of distractors compared to training time as our
> > > novel setting.
> >
> > Similar rewritings would benefit the rest of the introduction.
> >
> > >> The contributions and hypotheses are clear at a basic level, but they are
> > >> not explained in enough detail to make clear what the criteria for success
> > >> are.
> >
> > > In all cases our measure of success is team performance during test time,
> > > where the team could either be two pre-trained agents or a pre-trained
> > > listener agent and a simulated English speaker. We believe this evaluation is
> > > standard in EC literature, but we have also clarified this point in the
> > > current version.
> >
> > By criterion of success, I do not just mean a metric but a particular
> > relationship between variables which can be measured and determined to be above
> > or below a certain threshold.  For example, if we are expecting to see
> > a positive correlation between informativeness and generalizability, then
> > a clear criteria success would be a positive correlation coefficient between
> > the metrics for informativeness and generalizability.  Thus, team performance
> > is probably the correct metric here, but what is the particular relationship we
> > expect to see with that metric and other variables and how can we empirically
> > determine if we observe that relationship.

---

> > > ### Author Response · Authors · 2022-11-19
> > > **Statistical tests; stylistic differences.**
> > >
> > > **Conceptual scope of the paper:** this seems to be the main reviewer’s concern and the reviewer’s grounds for not recommending acceptance. While we agree with the reviewer that papers that consider “small but well-investigated claims” are valuable, we also believe that the community will greatly benefit from considering various types of papers, rather than just the one type suggested by the reviewer. This includes papers such as ours, which make broader conceptual contributions that are supported by a suite of carefully designed experiments, even (and perhaps especially) if these contributions could be further extended in future work.
> > >
> > > **Criterion of success:**
> > >
> > > > “By criterion of success, I do not just mean a metric but a particular relationship between variables which can be measured and determined to be above or below a certain threshold. For example, if we are expecting to see a positive correlation between informativeness and generalizability, then a clear criteria success would be a positive correlation coefficient between the metrics for informativeness and generalizability. Thus, team performance is probably the correct metric here, but what is the particular relationship we expect to see with that metric and other variables and how can we empirically determine if we observe that relationship”
> > >
> > > We are happy that the reviewer agrees with our choice of metric for evaluation, and we would like to note that the paper makes statements along the lines of what the reviewer suggests. For example, in Section 6, we write “[W]e expected that increasing $\lambda_I$ in training would improve the agents’ ability to generalize to OOD inputs.” This is a prediction about a general relationship that should hold between two well-defined variables. We then present data that directly supports this prediction (e.g., Figure 5, which includes error bars). Similarly, we state in our abstract that “when translating between English and EC, greater complexity leads to improved performance of teams of simulated English speakers and trained VQ-VIB listeners, but only up to a threshold corresponding to the English complexity.” This statement, relating two variables, is backed up by the evidence presented in the rest of the paper.
> > >
> > > In an attempt to further address the reviewer’s concern and reinforce the trends already apparent in our figures, we ran statistical analyses on the data in Figure 5 a (generalization to OOD inputs) and Figure 7 (translation utility).
> > >
> > > We performed ANOVA analysis for each of the three curves in Figure 5 a for C=2, 16, and 32. The estimated slopes for OOD utility vs. complexity for each curve were 0.09, 0.17, and 0.16, respectively, and, with a null hypothesis of zero slope, we found $p << 0.05$ ($t =4.6, 8.7, 9.6$) respectively. This establishes statistical significance for our hypothesized relationship between complexity and utility.
> > >
> > > For the translation experiment data in Figure 7, we compared nested models: one for a linear effect of complexity on translation utility for each $C$, and one with a pair of linear models of translation utility vs. complexity, fitted separately for complexity values greater than or less than 2 nats. A likelihood ratio test between these two models was significant at $p< 0.005$, indicating that there was indeed a plateauing behavior near the English complexity level.
> > > We thank the reviewer for the feedback on this point and think that these additional statistical analyses, while expected, do indeed strengthen the validity of our results. We have reported these statistical tests in the current revision of the paper (see Sections 6.2 and 6.3).
> > >
> > > **Organization and writing style:** We wish to note that all other reviewers found the paper well-written and clear, and therefore we feel that the reviewer’s rewriting suggestions may be mainly a matter of different stylistic tastes. We adopted a writing approach that starts from a high-level concise description in the abstract and introduction, and provides the full details and precise definitions later in the paper after the reader has some general idea of what will be discussed. Having said that, we would have been open to adjusting the wording along the lines of what the reviewer suggested; however, due to space limitations we were unable to expand the introduction and abstract with more technical details.

---

### Official Review · Reviewer_cAgg · 2022-10-26

**Confidence:** 4
**Correctness:** 3
**Technical Novelty And Significance:** 3
**Empirical Novelty And Significance:** 3
**Recommendation:** 8

**Clarity, Quality, Novelty And Reproducibility:**

The paper is clearly written, the experiments are high-quality and interesting. The authors shared the code.

**Strength And Weaknesses:**

Strengths:
* The paper is clearly written and well-motivated, the findings are novel;
* The experiments are convincing (taking into account Appendix C);
* The simulated English translation task is very interesting.

Weaknessess (mostly minor):
* I believe the abstract can be more specific and indicate that the "human-agent interaction" is "simulated human-agent interaction" and "English" is eg "English-based tags system";
* I am curious if there is a way for controlling the IB trade-off w/o imposing such strong architecture constraints. Eg, much of prior work operated on more standard architectures containing LSTM/GRU/Transformer blocks and producing sequentially generated multi-symbol messages. Are those compatible?
* The last paragraph of S6.2 mentions that using more candidates at training-time also allows better generalization - however, at the expense of lower performance, e.g. training with C=16 is twice faster than training with C=32. Is it possible to elaborate on where the difference is coming from? Since the image encoder is pretrained and frozen, the image embeddings can be precalculated and the difference between selecting between 16 or 32 candidates should be small comparing to the rest of the computations?
* While controlling the trade-off coefficient might be more efficient than increasing the number of candidates, the latter approach is universally applicable to any agent architecture. Is this worth highlighting?
* I believe that the proposed adaptation of VQ-VIB to producing multiple tokens independently is akin to the widely used product quantization scheme (Product Quantization for Nearest Neighbor Search, Jégou et al; VQ-WAV2VEC: Self-supervised Learning of Discrete Speech Representations, Baevski et al). Perhaps it is worth having a reference.

**Summary Of The Paper:**

This paper shows that explicitly controlling the trade-off between complexity, informativeness, and game utility in the emergent communication setup allows agents to discover languages with better generalization to harder/OOD tasks and better alignment with natural language. In order to demonstrate that, the authors study performance of a multi-symbol modification of VQ-VIB (which allows to control this trade-off) in a series of experiments, focusing on two setups: (a) referential games, and (b) human-agent translation.

The experiments performed on the ManyNames dataset demonstrate that:
* higher communication complexity correlates with higher utility;
* higher communication complexity correlates with better generalization to harder (more distractors) and OOD (unseen categories) settings;
* up to a bound, increasing communication complexity allows higher utility in a simulated translation task.

**Summary Of The Review:**

A very nice paper with interesting findings and high-quality experiments.

---

> ### Author Response · Authors · 2022-11-15
> **Minor clarifications, thanks for the references**
>
> We thank the reviewer for the very helpful comments and encouraging review!
> Below are our responses to the reviewer’s minor concerns (in the order in which they were listed by the reviewer):
>
> 1. **Abstract:**
> We absolutely agree and have updated the abstract accordingly.
>
> 2. **Architecture constraints:** This is an exciting direction that we look forward to exploring in the future. [Tucker et al. 2022](https://openreview.net/pdf?id=O5arhQvBdH) showed that some existing architectures (e.g., gumbel softmax) can be adapted to fit within the ITEC framework, and we corroborated those findings in this work. Further extensions to handle recurrent messages generated across timesteps would require some additional design choices but in principle, penalizing complexity with IB constraints should still apply. At the same time, we believe that the VQ-VIB method is currently one of the architectures best-suited towards the ITEC framework, through its complexity-limited representations in a continuous space.
>
>
> 3. **Timing experiments:** The reviewer suggested pre-calculating and then caching the extracted features for each image. Indeed, in order to create the fairest baselines, this is exactly what we did when generating the results for the paper. More specifically, at the start of every training run, we load the extracted features for every single image up front, and then just load data from that feature dataset. We have added a sentence noting this to Appendix A.
>
> &nbsp;&nbsp;&nbsp;&nbsp;&nbsp;&nbsp; In response to the reviewer’s questions, we timed some of the methods in our code and found that the greatest difference in timing occurred in loading each batch during training. Even though we pre-calculated image features, it was necessary to actually load such features as tensors on the GPU. The dataset was too large to pre-load everything onto our single GPU. Other aspects of the code (e.g., forward and backward passes of the agents) took similar amounts of time across different $C$.
>
>
> 4. **Comparisons to prior art:** Indeed, the reviewer makes a good point that prior work  ([Chaabouni et al. 2021](https://openreview.net/pdf?id=AUGBfDIV9rL)) that varies the training environment is somewhat architecture-agnostic (although agents must adapt to changes in input sizes) and therefore more general in that sense. We have sought to emphasize the generality of  [Chaabouni et al. 2021](https://openreview.net/pdf?id=AUGBfDIV9rL) in Section 6.2 in the latest revision. Ultimately, we view  [Chaabouni et al. 2021](https://openreview.net/pdf?id=AUGBfDIV9rL) and our work as complementary: is emergent communication guided by environmental (i.e., task-based) or intrinsic (e.g., IB terms) pressures? Likely, the answer is a mix of both, and we look forward to further investigations at the intersection of these approaches.
>
> 5. **Product Quantization references:**
> We thank the reviewer for including relevant references to Product Quantization work. We were unaware of such literature and have cited these works when introducing combinatorial VQ-VIB (Section 4.1).

---

### Official Review · Reviewer_3hpH · 2022-10-29

**Confidence:** 3
**Correctness:** 2
**Technical Novelty And Significance:** 2
**Empirical Novelty And Significance:** 2
**Recommendation:** 5

**Clarity, Quality, Novelty And Reproducibility:**

The presentation is generally well-written but the formalization is less so, and there is a lot of abuse of notation (though this does not to my knowledge hinder readability).  Reproducibility would be difficult from the paper alone, however the authors intend to release their code.  Novelty in modeling is limited.  The experimental design is quite novel, but difficult to understand the full consequences of this design and the implications of its findings on EC more generally.

**Strength And Weaknesses:**

The main contributions of this work are likely (1) the extensions to VQ-VIB, and (2) a sort of evaluation paradigm that involves combining learned EQ representations with word embeddings into the same communication game.  The VQ-VIB extension, generalizing the model from a single "word" to multiple words, is straightforward, but important.  It is easy to imagine how this could lead to more direct comparisons with traditional EC models, or how the decoder could been sequential instead of fully connected, such that there would have been more incentive for the model to learn a more compositional protocol.  At any rate, it is a necessary extension for moving towards any real form of human-like EC, where a message is composed of multiple words, where meaning is conveyed not just in the words chosen but by their order.


The second contribution, the experimental setup, is harder to evaluate because there are so many unorthodox choices going on, and there is not consensus in what assumptions are aligned with the goals of EC (or constitute EC).  In my mind, many of these choices make this work more about grounding between two models, or more like unsupervised machine translation, than proper EC.  This makes it hard for me to evaluate the proposed findings as such.

Another comment on the modeling -- I was surprised to see the decoder is trained in a reconstruction style objective using an MSE loss, rather downstream from the classification loss on the signaling game.  What was the motivation here?  Is it reasonable in an EC setting to assume a connection between the speaker observation and the decoder on the other side of a learned protocol?  Doesn't this reduce the EC task substantially?

I found some of the formal descriptions to be a bit messy.  m as a meaning and not a message / c as a communication and not a context/content, as is the case for many EC presentations, was slightly disorienting.  The KL is denoted with D and not KL.  Then D serves double duty as the decoder.  The lines are blurred between minimizing objectives over single m/c and the set of Ms/Cs (not defined).  Then c does double duty as indexes into the set of candidates.  Overall there is a lot of poorly chosen terminology in Section 3.

I would also argue that some of the preliminary discussion dealing with the complexity of the English naming system have less to do with a general property of English, and more specifically with properties of the dataset and the granularity of the naming used there.  It was not clear to me if the authors agree with this point, since some of the latter discussion then seemed to explicitly mention the dataset when talking about this point.

**Summary Of The Paper:**

The authors propose some extensions to existing work on VQ-VIB architecture for modeling EC with a more explicitly information theoretic framework.  Namely, they extend the number of tokens which are chosen, and in doing so combine two threads of EC research -- existing VQ-VIB, and more traditional RNN/sequential type architectures.  In a set of experiments on English, the authors vary aspects of the model such as informativeness, showing among other things that increased informativeness is associated with better test time generalization, up until the point where it matches or exceeds that of the communication partner's language.

**Summary Of The Review:**

Overall, I am still undecided on the overall merit of this paper.  On one hand, there is modest modeling contributions, and the experimental setup is unlike any I recall seeing before in EC.  The connections to the information theoretic complexity of English (of this dataset) is surprising in that it closely approximates previous attempts to calculate this number.

On the other hand, the formal presentation is messy, and the experiment setup seems to deviate strongly from standard EC assumptions.  Is it EC if one half of the protocol is derived from word embeddings of image labels?  Or if direct supervision is injected between the speaker and decoded output?  I have a hard time finding takeaway lessons that are both nonobvious and apply to the field more broadly.  The discussion contained within the paper does not go a long way towards reconciling this design with most existing EC work.  Similarly, this task seems setup specifically towards noun communication, but to what extent would many of these experiment choices work on a more general task with multiple parts-of-speech?

At a minimum, for a paper claiming that most existing EC methods fall short of achieving desired properties in ways that this work presumably does not, it could obviously benefit from direct comparison with such methods.  Do existing works not also manipulate the same factors?  Informativeness is increased here through means like scaling factors on loss terms or increases in the vocab size and number of words communicated, but existing work also explores the effect of varying such things, albeit more removed from a formal information theoretic formulation.  I'm not sure a comparison to prototype style EC is sufficiently representative of EC methods in general (conclusions are phrased wrt to these more general methods)

---

> ### Author Response · Authors · 2022-11-15
> **Response part 1: overview**
>
> Thank you for these valuable and thoughtful comments. Before we address all the reviewer’s concerns, we would like to clarify the key contributions of the paper, as we feel that some important components have been left out in this review. As summarized in the paper’s contributions section (Section 7), we show that: “(1) encouraging informativeness allows agents to better generalize to more challenging tasks and out-of-distribution inputs; (2) the structure of the emergent VQ-VIB communication vectors encodes some semantic similarities and facilitates open-domain communication, similar to word embeddings in natural language; and (3) performance for teams of simulated English speakers and trained EC listeners improves with the complexity of the EC system, but only up to the complexity level of the English naming system. These results suggest that taking into account the IB informativeness-complexity tradeoff in EC, in addition to maximizing utility, may support both self-play performance and human-agent interaction.”
>
> As noted by the reviewer, the extension of the VQ-VIB architecture is indeed another contribution of the paper that is designed to support future extensions of this framework. However, the second contribution listed by the reviewer (“a sort of evaluation paradigm that involves combining learned EQ representations with word embeddings into the same communication game”) only partially captures one of our experiments. The reviewer then raises the following concern about that aspect of the work:
>
> > “The second contribution, the experimental setup, is harder to evaluate because there are so many unorthodox choices going on, and there is not consensus in what assumptions are aligned with the goals of EC (or constitute EC). In my mind, many of these choices make this work more about grounding between two models, or more like unsupervised machine translation, than proper EC. This makes it hard for me to evaluate the proposed findings as such.”
>
> We wish to clarify that in all of our experiments, agents are trained in reference games with no English words or word embeddings as inputs. This is a standard EC approach, where agents are trained in “self-play” ([Sukhbaatar et al. 2016](https://proceedings.neurips.cc/paper/2016/file/55b1927fdafef39c48e5b73b5d61ea60-Paper.pdf) and [Lazaridou et al. 2017](https://openreview.net/pdf?id=Hk8N3Sclg) for normal self-play, or [Lowe et al. 2020](https://www.researchgate.net/publication/339041316_On_the_interaction_between_supervision_and_self-play_in_emergent_communication) for explicitly considering the interaction between self-play and supervision). After training, the agents’ weights are frozen. Our first set of experiments are not related to translation, but rather to testing the ability of our pre-trained agents to generalize to harder tasks and out-of-domain inputs (Figure 5 and Table 1). This builds on the recently proposed benchmark of [Chaabouni et al. 2021](https://openreview.net/pdf?id=AUGBfDIV9rL), measuring generalization to more distractors and out-of-distribution inputs. Only in our last experiment, which tests how well our pre-trained agents can interact with a simulated English speaker, did we use a very small amount of supervised data to learn a mapping (“translator”) between English word embeddings and the (pre-trained) emergent communication signals of our agents. It is hard to imagine how one might test human-agent interaction in EC settings without learning such a mapping.
>
> Given that we have used standard practices in machine learning, and in EC in particular, we are unsure what are the “unorthodox choices” that the reviewer is referring to. We would also like to clarify that this work does not aim to address the question of what are the goals of EC, but rather defines very concrete benchmarks for evaluation. We believe that these benchmarks – generalization to harder tasks, out-of-distribution generalization, and successful communication with humans – capture common intuitions about useful EC that would be valuable for the broader ICLR community, but we acknowledge that these intuitions may not be shared by everyone and our work does not rule out other types of benchmarks.

---

> > ### Author Response · Authors · 2022-11-15
> > **Response part 2**
> >
> > > “I was surprised to see the decoder is trained in a reconstruction style objective using an MSE loss, rather downstream from the classification loss on the signaling game. What was the motivation here? Is it reasonable in an EC setting to assume a connection between the speaker observation and the decoder on the other side of a learned protocol? Doesn't this reduce the EC task substantially”
> >
> > We would like to note that the downstream classification loss in signaling games (as well as the reconstruction loss) also depends on the speaker’s observation, as both agents are rewarded based on the match between the speaker’s target and the listener’s decision. Therefore, the reconstruction loss does not make different assumptions about the information available to the two agents compared to training with the classification loss, which is standard in EC. In both cases, one natural way to interpret this is by assuming that in training, after each episode, the speaker’s target and listener’s decision are revealed to both agents.
> >
> > More specifically, using a decoder for informative communication was previously proposed and justified by [Tucker et al. 2022](https://openreview.net/pdf?id=O5arhQvBdH)  (and see [Lin et al. 2021](https://toruowo.github.io/marl-ae-comm/resources/marl-ae-comm.pdf) for a related autoencoding approach). Intuitively, the motivation comes from thinking of a listener’s decision process as based on two steps: first, understand the speaker’s communication regardless of any specific task (hence, the reconstruction objective), and second, make a decision based on that understanding (and possibly additional observations). For example, if two humans were playing a reference game and a listener heard the word “duck”, they might first think of what ducks look like (decoding the communication) and then look for a photo of the duck (making a decision of the target image). In our implementation, these two steps gave rise to the decoder network, $D$, and the listener network, $L$, respectively (see Figure 1). Indeed, this general framework of decoding communication is motivated from prior literature studying human naming systems ([Zaslavsky et al. 2018](https://www.pnas.org/doi/10.1073/pnas.1800521115)).
> >
> >
> > > “I found some of the formal descriptions to be a bit messy. m as a meaning and not a message / c as a communication and not a context/content, as is the case for many EC presentations, was slightly disorienting. The KL is denoted with D and not KL. Then D serves double duty as the decoder. The lines are blurred between minimizing objectives over single m/c and the set of Ms/Cs (not defined). Then c does double duty as indexes into the set of candidates. Overall there is a lot of poorly chosen terminology in Section 3.”
> >
> > Thank you for this helpful feedback. While indeed, some papers in EC use different notations, we have decided to adopt the notation from the prior literature that our work directly builds on. The meaning $m$ can also be considered as an “intended message”, and we believe this is quite standard. While the notation $c$ for a communication vector has been used in prior work, we agree that it is less standard and might be confusing. We are currently working on changing it to $w$, which is a standard notation for a vector as well as for a word, and will soon upload a revision with this change. We also agree with the comment about $D$ and in the latest revision we uploaded, we have explicitly marked all KL divergences as $D_{KL}$. As for the minimization objective, we have adopted a standard convention to simplify notation by using lower case letters to denote both random variables and their instantiation. The minimization is not over single instances but rather over the parameters of their distributions. We believe that a possible source of confusion is the fact that in our initial submission there was a missing expectation sign in Equation 1, which we have now fixed. As for the definitions of the sets, we noted in section 3.2 that in our model, $m$ is an output of an image VAE, but following this reviewer’s comment, in the current version we state more clearly that $m\in\mathbb{R}^d$. Lastly, the set of communication vectors is not fixed, but rather trained, which is a key feature of VQ-VIB, as explained in section 3.2 (“... looking up the nearest element of a (trainable) codebook of $k$ quantization vectors $\zeta\in \mathbb{R}^Z$. The final communication vector output by the speaker, $c$, is this nearest quantization vector.”).

---

> > > ### Author Response · Authors · 2022-11-15
> > > **Response part 3:**
> > >
> > > >“I would also argue that some of the preliminary discussion dealing with the complexity of the English naming system have less to do with a general property of English, and more specifically with properties of the dataset and the granularity of the naming used there. It was not clear to me if the authors agree with this point, since some of the latter discussion then seemed to explicitly mention the dataset when talking about this point.”
> > >
> > >
> > > In this work, we measured the complexity of the English words associated with the stimuli set of the ManyNames dataset we used (e.g., green line in Figure 7). While this is indeed a property of stimuli set, it is also a property of the English language. We specifically chose the ManyNames dataset because, in contrast to image classification datasets that use simplified labels, the ManyNames dataset was constructed using a free-naming experiment with approximately 36 native English speakers for each image. This connects our work to the broader literature that considers human naming systems for a wide variety of languages and semantic domains and measures the complexity of each language from naming data (e.g. [Zaslavsky et al. 2018](https://www.pnas.org/doi/10.1073/pnas.1800521115) and [Zaslavsky et al. 2019](https://arxiv.org/abs/1905.04562)). We do not argue that our measure of complexity corresponds to the complexity of English as a whole, but rather to the complexity of the English naming system for visual objects (now highlighted in Section 6.3 of our revision).
> > >
> > > > “Reproducibility would be difficult from the paper alone, however the authors intend to release their code. Novelty in modeling is limited. The experimental design is quite novel, but difficult to understand the full consequences of this design and the implications of its findings on EC more generally.”
> > >
> > > We wish to note that all our implementational details are provided in the appendix, due to the space limitations for the main text. We are therefore unsure which details are missing to allow reproducibility from the paper alone, and would be grateful if the reviewer could be more specific about this. We also wish to note that we have already made our code available in the initial submission via this anonymous link: https://anonymous.4open.science/r/iclr_vqvib-B185/README.md
> > >
> > > As for the reviewer’s concern about the broader consequences and implications for EC research, we hope that our clarifications above regarding the paper’s contributions have addressed this matter. We are of course happy to address any further questions that the reviewer may have.

---

> > > > ### Author Response · Authors · 2022-11-15
> > > > **Response part 4:**
> > > >
> > > > > “Overall, I am still undecided on the overall merit of this paper. On one hand, there is modest modeling contributions, and the experimental setup is unlike any I recall seeing before in EC. The connections to the information theoretic complexity of English (of this dataset) is surprising in that it closely approximates previous attempts to calculate this number.
> > > > On the other hand, the formal presentation is messy, and the experiment setup seems to deviate strongly from standard EC assumptions. Is it EC if one half of the protocol is derived from word embeddings of image labels? Or if direct supervision is injected between the speaker and decoded output? I have a hard time finding takeaway lessons that are both nonobvious and apply to the field more broadly. The discussion contained within the paper does not go a long way towards reconciling this design with most existing EC work”
> > > >
> > > > We are very happy that the reviewer finds our experimental setup novel and the approximation of the English complexity interesting. More generally, our work is part of a new research area that connects EC with both information-theoretic principles and semantic embedding spaces (which are similar to word embeddings), and we are very excited to share our results in this area. While our experimental setup is novel, we do not think it deviates from any standard EC assumptions. As we clarified above, it is incorrect that “half of the protocol is derived from word embeddings of image labels.” Our agents are never trained with such data, but only via standard unsupervised self-play protocols. Furthermore, our set of experiments that test generalization to harder tasks and out-of-distribution inputs do not involve English word embeddings at all. Only our last experiment, in which we tested how well our pre-trained agents may communicate with simulated English speakers, required a very small amount of supervision data (N=100 for results in the paper, with other N reported in Appendix E, Figure 12) only to translate between the emergent (unsupervised) communication signals and English word embeddings. This translation is not intrinsic to our agents and does not affect in any way their weights or communication system. We find it quite remarkable and non-trivial that it is possible to map so easily between our EC systems and English word embeddings.
> > > >
> > > > The takeaways from our experiments are that our VQ-VIB agents learn to coordinate EC systems that generalize to harder tasks, to out-of-distribution inputs, and support human-agent communication, better than other standard baselines. In addition, our results suggest that the emergent VQ-VIB semantic space (i.e., the trainable codebook of discrete signals which are embedded in a continuous space) may explain why our agents achieve this remarkable performance in our downstream tasks. We believe that these contributions are novel and would be of broad interest to anyone who is interested in novel approaches to representation learning.
> > > >
> > > > > “This task seems setup specifically towards noun communication, but to what extent would many of these experiment choices work on a more general task with multiple parts-of-speech”
> > > >
> > > > While we have indeed considered only noun communication in this work, there are several reasons to believe that the same approach would also apply to other parts of speech. First, [Tucker et al. 2022](https://openreview.net/pdf?id=O5arhQvBdH)  have applied VQ-VIB to adjective communication (colors) and to navigation tasks in which one agent “commands” another (e.g., “go to location x”) which could be considered as a step toward verb-like communication. Second, the Information Bottleneck for human semantic systems, which our framework extends, has been applied not only to objects and colors, but also to function words (pronouns, [Zaslavsky et al. 2021](https://www.researchgate.net/publication/351563287_Let%27s_talk_efficiently_about_us_Person_systems_achieve_near-optimal_compression)) and grammatical markers ([Mollica et al. 2021](https://www.pnas.org/doi/10.1073/pnas.2025993118)). Third, as the reviewer notes, our extension of the VQ-VIB architecture enables more complex communication, which could be used in future work to further extend our approach to sequences of signals that may correspond to multiple parts of speech. We are very excited to explore these extensions in future work and have included a note to this effect in Section 7 of the latest revision.

---

> > > > > ### Author Response · Authors · 2022-11-15
> > > > > **Response part 5**
> > > > >
> > > > > > “At a minimum, for a paper claiming that most existing EC methods fall short of achieving desired properties in ways that this work presumably does not, it could obviously benefit from direct comparison with such methods. Do existing works not also manipulate the same factors? Informativeness is increased here through means like scaling factors on loss terms or increases in the vocab size and number of words communicated, but existing work also explores the effect of varying such things, albeit more removed from a formal information theoretic formulation. I'm not sure a comparison to prototype style EC is sufficiently representative of EC methods in general (conclusions are phrased wrt to these more general methods)”
> > > > >
> > > > > First, we wish to emphasize that we have done an extensive comparison with previous methods. Due to space constraints, in our initial submission these results were included in Appendix C, but we agree that this should be highlighted also in the main text. In the current revision we added Figure 5c, which summarizes the main comparison with baselines (the full set of results is still in Appendix C). Please note that we have not only compared with prototype style EC but also with the very popular one-hot method.
> > > > >
> > > > >
> > > > > To the best of our knowledge, no existing works besides [Tucker et al. 2022](https://openreview.net/pdf?id=O5arhQvBdH) explicitly consider the role of varying an informativeness loss weight in training EC agents. [Lin et al. 2021](https://toruowo.github.io/marl-ae-comm/resources/marl-ae-comm.pdf) used a reconstruction loss to generate communication, but they struggled to combine reconstruction and utility losses to guide emergent communication (see their Section 5.4 where they state “the model trained jointly with reinforcement learning [, combining utility and reconstruction,] consistently performed worse [than only trained with reconstruction losses]”). Their approach would be similar to if we set $\lambda_I=1$ and $\lambda_U=\lambda_C=0$. [Tucker et al. 2022](https://openreview.net/pdf?id=O5arhQvBdH) traded off terms for utility, informativeness, and complexity in guiding emergent communication, and found that encouraging informativeness induced faster convergence and higher utility. However, they primarily varied $\lambda_C$, the weight for complexity, rather than varying $\lambda_I$. Compared to their work, we explored the role of informativeness in more depth, with experiments designed specifically to differentiate between informativeness and utility. In [Tucker et al. 2022](https://openreview.net/pdf?id=O5arhQvBdH)’s experiments, both informativeness and utility were maximized by the same policy. In our experiments, we used reference games with carefully chosen candidate images to make the utility and informativeness terms distinct (see Section 5, subsection “Training setup”). Furthermore, we more carefully considered the effect of tuning $\lambda_I$ on aspects of team performance beyond self-play. [Tucker et al. 2022](https://openreview.net/pdf?id=O5arhQvBdH)  tested three settings for $\lambda_I$ and concluded that “using a higher weight for informativeness ($\lambda_I$) resulted in faster convergence to higher rewards” but only evaluated agents in the same sorts of environments as used during training. We evaluated for a greater range of $\lambda_I$ and connected informativeness to out-of-distribution and translation performance, which were not studied in [Tucker et al. 2022](https://openreview.net/pdf?id=O5arhQvBdH) .
> > > > >
> > > > > The reviewer asked if we compared to other EC methods that indirectly influence the informativeness of communication. We note that, while our main paper focuses on VQ-VIB, we compare architectures by testing Prototype and onehot (often the standard) architectures. Results from such comparisons are in Appendix C (e.g., see Table 2, and Figure 9). In response to this reviewer’s questions, we conducted further experiments with onehot agents parametrized with even more tokens (2048) but found no significant change in results. Beyond architectural changes, one can view some aspects of prior EC work as indirectly influencing the informativeness of communication. For example, we compare to [Chaabouni et al. 2021](https://openreview.net/pdf?id=AUGBfDIV9rL) as a baseline in which agents are trained in reference games with many candidate images. Such training appears to induce more complex and informative communication (see “Comp.” column in Table 1). This task-based approach is interesting because it shows how environmental factors can influence information-theoretic properties of EC indirectly, but, perhaps unsurprisingly, we find that directly optimizing to vary informativeness and complexity can induce greater effects.

---

### Author Response · Authors · 2022-11-15
**Overall interest among reviewers; highlighting comparisons to baselines**

We thank all reviewers for their careful consideration of this work and valuable feedback. We are extremely happy that all the reviewers found our work interesting and relevant to the field.

We have addressed specific reviewer questions/concerns in direct response to their comments, but here we would like to highlight one common theme of our responses. Several reviewers asked about comparisons to baselines, including other agent architectures (YuEu specifically, and QWC4 asked for more extensive experiments in general). We had included results for comparisons to other agents architectures in Appendix C in the original submission, and given the common interest among reviewers in surfacing such data, we have included some of these results in Section 6 of the main text in our latest revision and more prominently noted that further results are available in appendices.

---

### Decision · Program_Chairs · 2023-01-20

**Decision:**

Reject

**Justification For Why Not Higher Score:**

There is one weak reject and one strong reject for the paper. By reading the paper by myself, I tend to agree with their raised weaknesses.  The two reviewers haven't been convinced by the rebuttal.

**Justification For Why Not Lower Score:**

N/A

**Metareview: Summary, Strengths And Weaknesses:**

There is one weak reject and one strong reject for the paper. By reading the paper by myself, I tend to agree with their raised weaknesses.  The two reviewers haven't been convinced by the rebuttal.

Moreover, besides the clarity/paper organization issues raised by the two reviewers, I found the technical contribution of the paper is limited. Firstly, the proposed extension to VQ-VIB is a minor contribution. As there exist many EC models that do not rely on VQ, this contribution only affect a small portion of EC research. Secondly, the contribution of using translation to analyze and evaluate emergent language is not novel (e.g., Yao et al., 2022. Linking Emergent and Natural Languages via Corpus Transfer). This limits the second technical contribution, and weakened Reviewer 2's (the one who champions the acceptance) claim of the paper's strength.

On the other hand, I do agree with the reviewers that the paper demonstrates novel findings, which is an important direction in the EC field. As mentioned by Reviewer 3 "the paper makes too many claims which means that each claim receives only a small amount of empirical evaluation, analysis, and discussion", which I agree, I think the paper can be made stronger to focus on the analysis. Hope the reviews are useful for the authors to improve the paper in the future.